# DIFFUSION MODELS LEARN LOW-DIMENSIONAL DISTRIBUTIONS VIA SUBSPACE CLUSTERING

## ABSTRACT

Recent empirical studies have demonstrated that diffusion models can effectively learn the image distribution and generate new samples. Remarkably, these models can achieve this even with a small number of training samples despite a large image dimension, circumventing the curse of dimensionality. In this work, we provide theoretical insights into this phenomenon by leveraging key empirical observations: (i) the low intrinsic dimensionality of image data, (ii) a union of manifold structure of image data, and (iii) the low-rank property of the denoising autoencoder in trained diffusion models. These observations motivate us to assume the underlying data distribution of image data as a mixture of low-rank Gaussians and to parameterize the denoising autoencoder as a low-rank model according to the score function of the assumed distribution. With these setups, we rigorously show that optimizing the training loss of diffusion models is equivalent to solving the canonical subspace clustering problem over the training samples. Based on this equivalence, we further show that the minimal number of samples required to learn the underlying distribution scales linearly with the intrinsic dimensions under the above data and model assumptions. This insight sheds light on why diffusion models can break the curse of dimensionality and exhibit the phase transition from failure to success in learning distributions. Moreover, we empirically establish a correspondence between the subspaces and the semantic representations of image data, facilitating image editing. We validate these results with extensive experimental results on both simulated distributions and image datasets.

## 1 INTRODUCTION

Generative modeling is a fundamental task in deep learning, which aims to learn a data distribution from training data to generate new samples. Recently, diffusion models have emerged as a new family of generative models, demonstrating remarkable performance across diverse domains, including image generation (Alkhouri et al., 2024; Ho et al., 2020; Rombach et al., 2022), video content generation (Bar-Tal et al., 2024; Ho et al., 2022), speech and audio synthesis (Kong et al., 2020; 2021), and solving inverse problem (Chung et al., 2022; Song et al., 2024). In general, diffusion models learn a data distribution from training samples through a process that imitates the non-equilibrium thermodynamic diffusion process (Ho et al., 2020; Sohl-Dickstein et al., 2015; Song et al., 2021). Specifically, the training and sampling of diffusion models involve two stages: (i) a forward diffusion process where Gaussian noise is incrementally added to training samples at each time step, and (ii) a backward sampling process where the noise is progressively removed through a neural network that is trained to approximate the score function at all time steps. As described in prior works (Hyvärinen & Dayan, 2005; Song et al., 2021), the generative capability of diffusion models lies in their ability to learn the *score function* of the data distribution, i.e., the gradient of the logarithm of the *probability density function* (*pdf*). We refer the reader to (Chen et al., 2024a; Croitoru et al., 2023; Yang et al., 2023) for a more comprehensive introduction and survey on diffusion models.

Despite the recent advances in understanding sampling convergence (Chen et al., 2023b; Lee et al., 2022; Li et al., 2023), distribution learning (Chen et al., 2023a; Oko et al., 2023), memorization (Gu et al., 2023; Somepalli et al., 2023; Wen et al., 2023; Zhang et al., 2024), and generalization (Kadkhodaie et al., 2023; Yoon et al., 2023; Zhang et al., 2023) of diffusion models, the fundamental working mechanisms remain poorly understood. One of the key questions is

*When and why can diffusion models learn the underlying data distribution without suffering from the curse of dimensionality?*

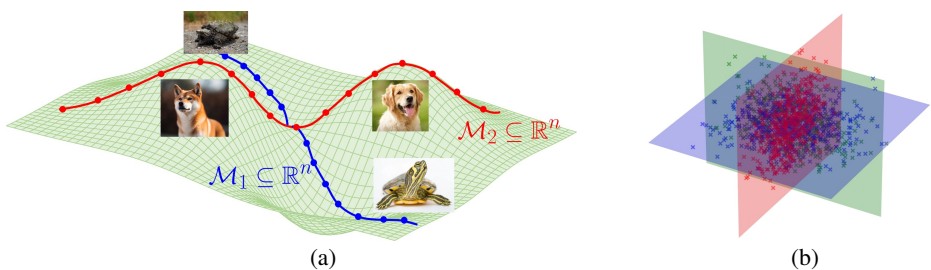

Figure 1: (a) Visualization of the union of manifold structure of image data. Here, different images lie on different manifolds $\mathcal{M}_i \subseteq \mathbb{R}^n$ of intrinsic dimension $d$ with $d \ll n$. (b) An illustration of training samples that are generated according to the MoLRG model. This model is a local linearization of a union of manifolds.

At first glance, the answer might seem quite straightforward. If a diffusion model can learn the empirical distribution of the training data that accurately approximates the underlying data distribution, then the puzzle is solved! However, it has been shown in (Li et al., 2024) that the number of samples for an empirical distribution to approximate the underlying data distribution could grow exponentially with respect to (*w.r.t.*) the data dimension. Moreover, Oko et al. (2023); Wibisono et al. (2024) showed that to learn an $\epsilon$-accurate score estimator measured by the $\ell_2$-norm via score matching or kernel-based approach, the required size of training samples grows at the rate of $O(\epsilon^{-n})$, where $n$ is the data dimension. These theoretical findings indicate that learning the underlying distribution via diffusion models suffers from the curse of dimensionality. In contrast, recent studies (Kadkhodaie et al., 2023; Zhang et al., 2023) showed that the number of training samples for a diffusion model to learn the underlying distribution is much *smaller* than the worst-case scenario, breaking the curse of dimensionality. Therefore, there is a significant gap between theory and practice.

In this work, we aim to address the above question of learning the underlying distribution via diffusion models by leveraging low-dimensional models. Our key observations are as follows: (i) The intrinsic dimensionality of real image data is significantly lower than the ambient dimension, a fact well-supported by extensive empirical evidence in Gong et al. (2019); Pope et al. (2020); Stanczuk et al. (2024); (ii) Image data lies on a disjoint union of manifolds of varying intrinsic dimensions, as empirically verified in Brown et al. (2023); Kamkari et al. (2024); Loaiza-Ganem et al. (2024) (see Figure 1(a)); (iii) We empirically observe that the denoising autoencoder (DAE) (Pretorius et al., 2018; Vincent, 2011) of diffusion models trained on real-world image datasets exhibit low-rank structures (see Figure 3). Based on these observations, we conduct a theoretical investigation of distribution learning through diffusion models by assuming that (i) the underlying data distribution is a *mixture of low-rank Gaussians* (see Definition 1) and (ii) the denoising autoencoder is parameterized according to the score function of the MoLRG. Notably, these assumptions will be carefully discussed based on the existing literature and validated by our experiments on real image datasets.

## 1.1 Our Contributions

This work studies the DAE-based training loss of diffusion models under the above low-dimensional data model and network parameterization. Our contributions can be summarized as follows:

- **Equivalence between training diffusion models and subspace clustering.** Under the above setup, we show that the training loss of diffusion models is equivalent to the *unsupervised* subspace clustering problem (Agarwal & Mustafa, 2004; Vidal, 2011; Wang et al., 2022) (see Theorem 3). This equivalence implies that training diffusion models is essentially learning low-dimensional manifolds of the data distribution.

- **Understanding breaking the curse of dimensionality in learning distributions.** By leveraging the above equivalence and the data model, we show that if the number of samples exceed the intrinsic dimension of the subspaces, the optimal solutions of the training loss can recover the underlying distribution. This explains why diffusion models can break the curse of dimensionality. Conversely, if the number of samples is insufficient, it may learn an incorrect distribution.

- **Correspondence between semantic representations and the subspaces.** Interestingly, we find that the discovered low-dimensional subspaces in a pre-trained diffusion model possess *semantic*

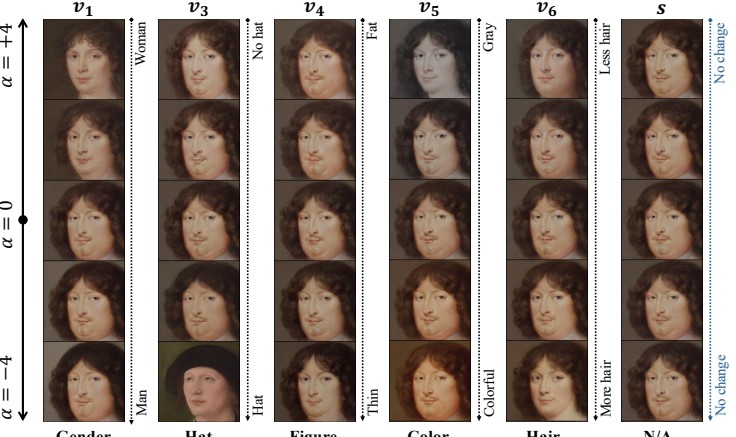

Figure 2: **Correspondence between the singular vectors of the Jacobian of the DAE and semantic image attributes.** We use a pre-trained DDPM with U-Net on the MetFaces dataset (Karras et al., 2020). We edit the original image $x_0$ by changing $x_t$ into $x_t + \alpha v_i$, where $v_i$ is a singular vector of the Jacobian of the DAE $x_\theta(x_t, t)$. In the last column, the editing direction $s$ is random.

meanings for natural images; see Figure 2. This motivates us to propose a training-free method to edit images on a frozen-trained diffusion model.

We also conduct extensive numerical experiments on both synthetic and real data sets to verify our assumptions and validate our theory. More broadly, the theoretical insights we gained in this work provide practical guidance as follows. First, we have shown that the number of samples for learning the underlying distribution via diffusion models scales proportionally with its intrinsic dimension. This insight allows us to improve training efficiency by quantifying the number of required training samples. Second, the identified correspondence between semantic representations and subspaces provides valuable guidance on controlling data generation. By manipulating the semantic representations within these subspaces, we can achieve more precise and targeted data generation.

### 1.2 RELATED WORKS AND DISCUSSIONS

**Learning a mixture of Gaussians via diffusion models.** Recent works have extensively studied distribution learning and generalizability of diffusion models for learning a mixture of full-rank Gaussian (MoG) model (Chen et al., 2024b; Cole & Lu, 2024; Gatmiry et al., 2024; Shah et al., 2023; Wu et al., 2024). Specifically, they assumed that there exist centers $\mu_1, \ldots, \mu_K \in \mathbb{R}^n$ such that image data approximately follows from the following distribution:

$$x \sim \sum_{k=1}^{K} \pi_k \mathcal{N}(\mu_k, I_n), \tag{1}$$

where $\pi_k \geq 0$ is the mixing proportion of the $k$-th mixture component satisfying $\sum_{k=1}^{K} \pi_k = 1$. Notably, the MoLRG model is distinct from the above MoG model that is widely studied in the literature. Specifically, the MoG model consists of multiple Gaussians with varying means and covariance spanning the full-dimensional space (see Eq. (1)), while a MoLRG comprises multiple Gaussians with zero mean and low-rank covariance (see Eq. (7)), lying in a union of low-dimensional subspaces. As such, the MoLRG model, inspired by the inherent low-dimensionality of image datasets (Gong et al., 2019; Pope et al., 2020; Stanczuk et al., 2024), offers a deeper insight into how diffusion models can learn underlying distributions in practice without suffering from the curse of dimensionality.

**Memorization and generalization in diffusion models.** Recently, extensive studies (Kadkhodaie et al., 2023; Yoon et al., 2023; Zhang et al., 2023) empirically revealed that diffusion models learn the score function across two distinct regimes — memorization (i.e., learning the empirical distribution) and generalization (i.e., learning the underlying distribution) — depending on the training dataset size vs. the model capacity. For a model with a fixed number of parameters, there is a phase transition from memorization to generalization as the number of training samples increases

(Kadkhodaie et al., 2023; Zhang et al., 2023). Notably, most existing studies on the memorization and generalization of diffusion models are empirical. In contrast, our work provides rigorous theoretical explanations for these intriguing experimental observations based on the MoLRG model. We demonstrate that diffusion models learn the underlying data distribution with the number of training samples scaling linearly with the intrinsic dimension.

**Notation.** We write matrices in bold capital letters like $\boldsymbol{A}$, vectors in bold lower-case letters like $\boldsymbol{a}$, and scalars in plain letters like $a$. Given a matrix, we use $\|\boldsymbol{A}\|$ to denote its largest singular value (i.e., spectral norm), $\sigma_i(\boldsymbol{A})$ its $i$-th largest singular value, and $a_{ij}$ its $(i,j)$-th entry, $\mathrm{rank}(\boldsymbol{A})$ its rank, $\|\boldsymbol{A}\|_F$ its Frobenius norm. Given a vector $\boldsymbol{a}$, we use $\|\boldsymbol{a}\|$ to denote its Euclidean norm and $a_i$ its $i$-th entry. Let $\mathcal{O}^{n \times d}$ denote the set of all $n \times d$ orthonromal matrices. We simply write the score function $\nabla_{\boldsymbol{x}} \log p(\boldsymbol{x})$ of a distribution with probability density function (pdf) $p(\boldsymbol{x})$ as $\nabla \log p(\boldsymbol{x})$.

## 2 PROBLEM SETUP

In this work, we consider an image dataset consisting of samples $\{\boldsymbol{x}^{(i)}\}_{i=1}^N \subseteq \mathbb{R}^n$, where each data point is *i.i.d.* sampled from an underlying data distribution $p_{\mathrm{data}}(\boldsymbol{x})$. Instead of learning this pdf directly, score-based diffusion models aim to learn the score function from the training samples.

### 2.1 PRELIMINARIES ON SCORE-BASED DIFFUSION MODELS

**Forward and reverse SDEs of diffusion models.** In general, diffusion models consist of forward and reverse processes indexed by a continuous time variable $t \in [0, 1]$. Specifically, the forward process progressively injects noise into the data. This process can be described by the following stochastic differential equation (SDE):

$$\mathrm{d}\boldsymbol{x}_t = f(t)\boldsymbol{x}_t \mathrm{d}t + g(t)\mathrm{d}\boldsymbol{w}_t, \tag{2}$$

where $\boldsymbol{x}_0 \sim p_{\mathrm{data}}$, the scalar functions $f(t), g(t) : \mathbb{R} \to \mathbb{R}$ respectively denote the drift and diffusion coefficients,[1] and $\{\boldsymbol{w}_t\}_{t \in [0,1]}$ is the standard Wiener process. For ease of exposition, let $p_t(\boldsymbol{x})$ denote the pdf of $\boldsymbol{x}_t$ and $p_t(\boldsymbol{x}_t|\boldsymbol{x}_0)$ the transition kernel from $\boldsymbol{x}_0$ to $\boldsymbol{x}_t$. According to Eq. (2), we have

$$p_t(\boldsymbol{x}_t|\boldsymbol{x}_0) = \mathcal{N}(\boldsymbol{x}_t; s_t\boldsymbol{x}_0, s_t^2\sigma_t^2\boldsymbol{I}_n), \text{ where } s_t = \exp\left(\int_0^t f(\xi)\mathrm{d}\xi\right), \sigma_t = \sqrt{\int_0^t \frac{g^2(\xi)}{s^2(\xi)}\mathrm{d}\xi}, \tag{3}$$

where $s_t := s(t)$ and $\sigma_t := \sigma(t)$ for simplicity. The reverse process gradually removes the noise from $\boldsymbol{x}_1$ via the following reverse-time SDE:

$$\mathrm{d}\boldsymbol{x}_t = \left(f(t)\boldsymbol{x}_t - g^2(t)\nabla \log p_t(\boldsymbol{x}_t)\right)\mathrm{d}t + g(t)\mathrm{d}\bar{\boldsymbol{w}}_t, \tag{4}$$

where $\{\bar{\boldsymbol{w}}_t\}_{t \in [0,1]}$ is another standard Wiener process, independent of $\{\boldsymbol{w}_t\}$, running backward in time from $t = 1$ to $t = 0$. It is worth noting that if $\boldsymbol{x}_1$ and $\nabla \log p_t$ are provided, the reverse process has exactly the same distribution as the forward process at each time $t \geq 0$ (Anderson, 1982).

**Training loss of diffusion models.** Unfortunately, the score function $\nabla \log p_t$ is usually unknown, as it depends on the underlying data distribution $p_{\mathrm{data}}$. To enable data generation via the reverse SDE (4), a common approach is to estimate the score function $\nabla \log p_t$ using the training samples $\{\boldsymbol{x}^{(i)}\}_{i=1}^N$ based on the scoring matching (Ho et al., 2020; Song et al., 2021). Because of the equivalence between the score function $\nabla \log p_t(\boldsymbol{x}_t)$ and the posterior mean $\mathbb{E}[\boldsymbol{x}_0|\boldsymbol{x}_t]$, i.e.,

$$s_t \mathbb{E}[\boldsymbol{x}_0|\boldsymbol{x}_t] = \boldsymbol{x}_t + s_t^2\sigma_t^2\nabla \log p_t(\boldsymbol{x}_t), \tag{5}$$

according to Tweedie's formula and (3), an alternative approach to estimate the score function $\nabla \log p_t$ is to estimate the posterior mean $\mathbb{E}[\boldsymbol{x}_0|\boldsymbol{x}_t]$. Consequently, extensive works (Chen et al., 2024c; Kadkhodaie et al., 2023; Karras et al., 2022; Vincent, 2011; Xiang et al., 2023) have considered training a time-dependent function $\boldsymbol{x}_{\boldsymbol{\theta}}(\cdot, t) : \mathbb{R}^n \times [0, 1] \to \mathbb{R}^n$, known as *denoising autoencoder* (DAE), parameterized by a neural network with parameters $\boldsymbol{\theta}$ to estimate the posterior mean $\mathbb{E}[\boldsymbol{x}_0|\boldsymbol{x}_t]$. To determine the parameters $\boldsymbol{\theta}$, we can minimize the following empirical loss:

$$\min_{\boldsymbol{\theta}} \ell(\boldsymbol{\theta}) := \frac{1}{N}\sum_{i=1}^N \int_0^1 \lambda_t \mathbb{E}_{\boldsymbol{\epsilon} \sim \mathcal{N}(\boldsymbol{0}, \boldsymbol{I}_n)}\left[\left\|\boldsymbol{x}_{\boldsymbol{\theta}}(s_t\boldsymbol{x}^{(i)} + \gamma_t\boldsymbol{\epsilon}, t) - \boldsymbol{x}^{(i)}\right\|^2\right]\mathrm{d}t, \tag{6}$$

---

[1]In general, the functions $f(t)$ and $g(t)$ are chosen such that (i) $\boldsymbol{x}_t$ for all $t$ close to 0 approximately follows the data distribution $p_{\mathrm{data}}$ and (ii) $\boldsymbol{x}_t$ for all $t$ close to 1 is nearly a standard Gaussian distribution; see, e.g., the settings in Ho et al. (2020); Karras et al. (2022); Song et al. (2021).

where $\lambda_t : [0, 1] \to \mathbb{R}^+$ is a weighting function and $\gamma_t := s_t \sigma_t$. As shown in Vincent (2011), training the DAE is equivalent to performing explicit or implicit score matching under mild conditions. We refer the reader to Appendix A.1 for the relationship between this loss and the score-matching loss in (Song et al., 2021; Vincent, 2011).

## 2.2 Low-Dimensional Data and Models

**Mixture of low-rank Gaussian data distribution.**    Although real-world image datasets are high dimensional in terms of pixel count and overall data volume, extensive empirical works (Gong et al., 2019; Kamkari et al., 2024; Pope et al., 2020; Stanczuk et al., 2024) suggest that their intrinsic dimensions are much lower. For instance, Pope et al. (2020) indicated that even for complex datasets like ImageNet (Russakovsky et al., 2015), the intrinsic dimensionality is approximately 40, which is significantly lower than its ambient dimension. Recently, Brown et al. (2023); Kamkari et al. (2024); Loaiza-Ganem et al. (2024) empirically validated the *union of manifolds* hypothesis, demonstrating that high-dimensional image data often lies on a disjoint union of manifolds instead of a single manifold. These observations motivate us to model the underlying data distribution as a *mixture of low-rank Gaussians*, where the data points are generated from a mixture of several Gaussian distributions; see Figure 1(b). We formally define the MoLRG distribution as follows:

**Definition 1** (Mixtures of Low-Rank Gaussians). *We say that a random vector $\boldsymbol{x} \in \mathbb{R}^n$ follows a mixture of $K$ low-rank Gaussian distribution with parameters $\{\pi_k\}_{k=1}^K$ and $\{\boldsymbol{U}_k^\star\}_{k=1}^K$ if*

$$\boldsymbol{x} \sim \sum_{k=1}^K \pi_k \mathcal{N}(\boldsymbol{0}, \boldsymbol{U}_k^\star \boldsymbol{U}_k^{\star T}), \tag{7}$$

*where $\boldsymbol{U}_k^\star \in \mathcal{O}^{n \times d_k}$ denotes the orthonormal base of the $k$-th component and $\pi_k \geq 0$ is the mixing proportion of the $k$-th mixture component satisfying $\sum_{k=1}^K \pi_k = 1$.*

Before we proceed, we make some remarks on this data model. First, to study how diffusion models learn the underlying data distribution, many recent works have studied a mixture of full-rank Gaussian distributions (see Eq. (1)); see, e.g., Chen et al. (2024b); Gatmiry et al. (2024); Shah et al. (2023). However, compared to this model, MoLRG is a more suitable model for capturing the low-dimensionality in image data distribution. Second, Brown et al. (2023); Kamkari et al. (2024) conducted extensive numerical experiments to validate that image datasets such as MNIST and ImageNet approximately lie on a union of low-dimensional manifolds. Because a nonlinear manifold can be well approximated by its tangent space (i.e., a linear subspace) in a local neighborhood, the MoLRG model, which represents data as a union of linear subspace, serves a good local approximation of a union of manifolds. Finally, assuming Gaussian distributions in each subspace in the MoLRG model is to ensure theoretical tractability, making it a practical starting point for theoretical studies on real-world image datasets. Now, we compute the ground-truth posterior mean $\mathbb{E}[\boldsymbol{x}_0 | \boldsymbol{x}_t]$ when $\boldsymbol{x}_0$ satisfies the MoLRG model as follows.

**Lemma 1.** *Suppose that $\boldsymbol{x}_0$ satisfies the MoLRG model. For each time $t > 0$, it holds that*

$$\mathbb{E}[\boldsymbol{x}_0 | \boldsymbol{x}_t] = \frac{s_t}{s_t^2 + \gamma_t^2} \frac{\sum_{k=1}^K \pi_k \exp\left(\phi_t \|\boldsymbol{U}_k^{\star T} \boldsymbol{x}_t\|^2\right) \boldsymbol{U}_k^\star \boldsymbol{U}_k^{\star T} \boldsymbol{x}_t}{\sum_{k=1}^K \pi_k \exp\left(\phi_t \|\boldsymbol{U}_k^{\star T} \boldsymbol{x}_t\|^2\right)}, \text{ where } \phi_t := \frac{s_t^2}{2\gamma_t^2(s_t^2 + \gamma_t^2)}. \tag{8}$$

We defer the proof of this lemma to Appendix A.2. Notably, this lemma provides guidance on the network parameterization of the DAE $\boldsymbol{x}_{\boldsymbol{\theta}}(\cdot, t)$ as discussed below.

**Low-rank network parameterization.**    When we train diffusion models with the U-Net architecture (Ronneberger et al., 2015) on various image datasets, it is observed that the numerical rank of the Jacobian of the DAE, i.e., $\nabla_{\boldsymbol{x}_t} \boldsymbol{x}_{\boldsymbol{\theta}}(\boldsymbol{x}_t, t)$, is substantially lower than the ambient dimension in most time steps; see Figure 3(a). When training diffusion models with U-Net on the samples generated according to the MoLRG model, the Jacobian of the DAE also exhibits a similar low-rank pattern; see Figure 3(b). The above observations motivate us to consider a low-rank parameterization of the network. According to the ground-truth posterior mean of the MoLRG model in Lemma 1, a natural parameterization for the DAE is

$$\boldsymbol{x}_{\boldsymbol{\theta}}(\boldsymbol{x}_t, t) = \frac{s_t}{s_t^2 + \gamma_t^2} \sum_{k=1}^K w_k(\boldsymbol{\theta}; \boldsymbol{x}_t) \boldsymbol{U}_k \boldsymbol{U}_k^T \boldsymbol{x}_t, \ w_k(\boldsymbol{\theta}; \boldsymbol{x}_t) = \frac{\pi_k \exp\left(\phi_t \|\boldsymbol{U}_k^T \boldsymbol{x}_t\|^2\right)}{\sum_{l=1}^K \pi_l \exp\left(\phi_t \|\boldsymbol{U}_l^T \boldsymbol{x}_t\|^2\right)}, \tag{9}$$

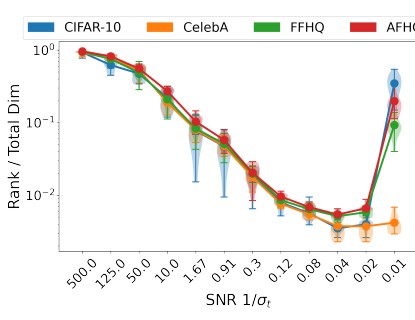
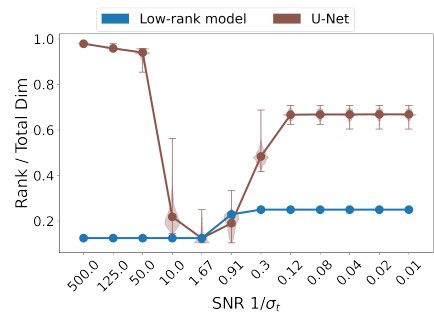

(a) **Real image datasets**  (b) **Mixture of low-rank Gaussians**

Figure 3: **Low-rank property of the DAE of trained diffusion models.** We plot the ratio of the numerical rank of the Jacobian of the denoising autoencoder, i.e., $\nabla_{\boldsymbol{x}_t}\boldsymbol{x}_\theta(\boldsymbol{x}_t, t)$, over the total dimension against the signal-to-noise ratio (SNR) $1/\sigma_t$ on trained diffusion models. (a) We train diffusion models on image datasets CIFAR-10, CelebA, FFHQ, and AFHQ. The experimental details are provided in Appendix C.1. (b) We respectively train diffusion models with the low-rank parameterization (9) and U-Net on a mixture of low-rank Gaussian distributions.

where the network parameters $\boldsymbol{\theta} = \{\boldsymbol{U}_k\}_{k=1}^K$ satisfy $\boldsymbol{U}_k \in \mathcal{O}^{n \times d_k}$. Although this approach may seem idealized, it offers several practical insights. First, if we consider a single low-rank Gaussian, the network parameterization takes the form $\boldsymbol{x} - s_t/(s_t^2 + \gamma_t^2)\boldsymbol{U}\boldsymbol{U}^T\boldsymbol{x}$, which resembles the structure of a practical U-Net with a linear encoder, decoder, and skip connections. This provides theoretical insights into why U-Net is preferred for training diffusion models. Second, to learn the underlying distribution, the number of samples should be proportional to its intrinsic dimension. In practice, this informs us on how to use a minimal number of samples to train diffusion models to achieve generalization.

Similar simplifications have been widely used for theoretical analysis in various ideal data distributions; see, e.g., Chen et al. (2023a; 2024b); Gatmiry et al. (2024); Shah et al. (2023). Notably, under this specific network parameterization in Eq. (9), learning the score function $\nabla \log p_t(\boldsymbol{x}_t)$ reduces to learning the network parameters $\boldsymbol{\theta}$ in Eq. (9) according to Lemma 1 and Eq. (5).

## 3 MAIN RESULTS

Based on the setups in Section 2.2, we are ready to conduct a theoretical analysis of distribution learning using diffusion models.

### 3.1 A WARM-UP STUDY: A SINGLE LOW-RANK GAUSSIAN CASE

To begin, we start from a simple case that the underlying distribution $p_{\text{data}}$ is a *single* low-rank Gaussian. Specifically, the training samples $\{\boldsymbol{x}^{(i)}\}_{i=1}^N \subseteq \mathbb{R}^n$ are generated according to

$$\boldsymbol{x}^{(i)} = \boldsymbol{U}^\star \boldsymbol{a}_i + \boldsymbol{e}_i, \tag{10}$$

where $\boldsymbol{U}^\star \in \mathcal{O}^{n \times d}$ denotes an orthonormal basis, $\boldsymbol{a}_i \overset{i.i.d.}{\sim} \mathcal{N}(\boldsymbol{0}, \boldsymbol{I}_d)$ is coefficients for each $i \in [N]$, and $\boldsymbol{e}_i \in \mathbb{R}^n$ is noise for all $i \in [N]$.[2] According to (9), we parameterize the DAE into

$$\boldsymbol{x}_\theta(\boldsymbol{x}_t, t) = \frac{s_t}{s_t^2 + \gamma_t^2}\boldsymbol{U}\boldsymbol{U}^T\boldsymbol{x}_t, \tag{11}$$

where $\boldsymbol{\theta} = \boldsymbol{U} \in \mathcal{O}^{n \times d}$. Equipped with the above setup, we can show the following result.

**Theorem 1.** *Suppose that the DAE $\boldsymbol{x}_\theta(\cdot, t)$ in Problem (6) is parameterized into (11) for each $t \in [0, 1]$. Then, Problem (6) is equivalent to the following PCA problem:*

$$\max_{\boldsymbol{U} \in \mathbb{R}^{n \times d}} \sum_{i=1}^N \|\boldsymbol{U}^T\boldsymbol{x}^{(i)}\|^2 \qquad \text{s.t.} \quad \boldsymbol{U}^T\boldsymbol{U} = \boldsymbol{I}_d. \tag{12}$$

---

[2]Since real-world images inherently contain noise due to various factors, such as sensor limitation, environment conditions, and transition error, it is reasonable to add a noise term to this model.

We defer the proof to Appendix A.3. In the single low-rank Gaussian model, Theorem 1 shows that training diffusion models with a DAE of the form (11) to learn this distribution is equivalent to performing PCA on the training samples. Leveraging this equivalence, we can further characterize the number of samples required for learning underlying distribution under the data model (10).

**Theorem 2.** *Consider the setting of Theorem 1. Suppose that the training samples $\{\boldsymbol{x}^{(i)}\}_{i=1}^N$ are generated according to the noisy single low-rank Gaussian model defined in (10). Let $\hat{\boldsymbol{U}}$ denote an optimal solution of Problem (6). The following statements hold:*

i) *If $N \geq d$, it holds with probability at least $1 - 1/2^{N-d+1} - \exp\left(-c_2 N\right)$ that any optimal solution $\hat{\boldsymbol{U}}$ satisfies*

$$\left\|\hat{\boldsymbol{U}}\hat{\boldsymbol{U}}^T - \boldsymbol{U}^\star \boldsymbol{U}^{\star T}\right\|_F \leq \frac{c_1 \sqrt{\sum_{i=1}^N \|\boldsymbol{e}_i\|^2}}{\sqrt{N} - \sqrt{d-1}}, \tag{13}$$

*where $c_1, c_2 > 0$ are constants that depend polynomially only on the Gaussian moment.*

ii) *If $N < d$, there exists an optimal solution $\hat{\boldsymbol{U}} \in \mathcal{O}^{n \times d}$ such that with probability at least $1 - 1/2^{d-N+1} - \exp\left(-c_2'd\right)$,*

$$\left\|\hat{\boldsymbol{U}}\hat{\boldsymbol{U}}^T - \boldsymbol{U}^\star \boldsymbol{U}^{\star T}\right\|_F \geq \sqrt{2\min\{d-N, n-d\}} - \frac{c_1'\sqrt{\sum_{i=1}^N \|\boldsymbol{e}_i\|^2}}{\sqrt{d} - \sqrt{N-1}}, \tag{14}$$

*where $c_1', c_2' > 0$ are constants that depend polynomially only on the Gaussian moment.*

**Remark 1.** We defer the proof to Appendix A.4. Building on the equivalence in Theorem 1 and the DAE parameterization (11), Theorem 2 clearly shows a phase transition from failure to success in learning the underlying distribution as the number of training samples increases. This phase transition is further corroborated by our experiments in Figures 4(a) and 4(b). Note that our theory cannot explain why diffusion models memorize training data (i.e., learning the empirical distribution). This is because the parameterization (11) is not as sufficiently over-parameterized as architectures like U-Net. We plan to explore this over-parameterized setting in future work to better understand how diffusion models achieve memorization and to extend our theoretical insights accordingly.

## 3.2 FROM SINGLE LOW-RANK GAUSSIAN TO MIXTURES OF LOW-RANK GAUSSIANS

In this subsection, we extend the above study to the MoLRG distribution. In particular, we consider a noisy version of the MoLRG model as defined Definition 1. Specifically, the training samples are generated by

$$\boldsymbol{x}^{(i)} = \boldsymbol{U}_k^\star \boldsymbol{a}_i + \boldsymbol{e}_i \text{ with probability } \pi_k, \quad \forall i \in [N], \tag{15}$$

where $\boldsymbol{U}_k^\star \in \mathcal{O}^{n \times d_k}$ denotes an orthonormal basis for each $k \in [K]$, $\boldsymbol{a}_i \overset{i.i.d.}{\sim} \mathcal{N}(\boldsymbol{0}, \boldsymbol{I}_{d_k})$ is coefficients, and $\boldsymbol{e}_i \in \mathbb{R}^n$ is noise for each $i \in [N]$. As argued by Brown et al. (2023), image data lies on a *disjoint* union of manifolds. This motivates us to assume that the basis matrices of subspaces satisfy $\boldsymbol{U}_k^{\star T}\boldsymbol{U}_l^\star = \boldsymbol{0}$ for each $k \neq l$. To simplify our analysis, we assume that $d_1 = \cdots = d_K = d$ and the mixing weights satisfy $\pi_1 = \cdots = \pi_K = 1/K$. Moreover, we consider a hard-max counterpart of Eq. (9) for the DAE parameterization as follows:

$$\boldsymbol{x}_{\boldsymbol{\theta}}(\boldsymbol{x}_t, t) = \frac{s_t}{s_t^2 + \gamma_t^2} \sum_{k=1}^K \hat{w}_k(\boldsymbol{\theta}, \boldsymbol{x}_0) \boldsymbol{U}_k \boldsymbol{U}_k^T \boldsymbol{x}_t, \tag{16}$$

where $\boldsymbol{\theta} = \{\boldsymbol{U}_k\}_{k=1}^K$ and the weights $\{\hat{w}_k(\boldsymbol{\theta}; \boldsymbol{x}_0)\}_{k=1}^K$ are set as

$$\hat{w}_k(\boldsymbol{\theta}; \boldsymbol{x}_0) = 1, \text{ if } k = k_0, \ \hat{w}_k(\boldsymbol{\theta}; \boldsymbol{x}_0) = 0, \text{ otherwise}, \tag{17}$$

where $k_0 \in [K]$ is an index satisfying $\|\boldsymbol{U}_{k_0}^T \boldsymbol{x}_0\| \geq \|\boldsymbol{U}_l^T \boldsymbol{x}_0\|$ for all $l \neq k_0 \in [K]$. We should point out that we use two key approximations here. First, the soft-max weights $\{w_k(\boldsymbol{\theta}, \boldsymbol{x}_t)\}$ in Eq. (9) are approximated by the hard-max weights $\{\hat{w}_k(\boldsymbol{\theta}; \boldsymbol{x}_0)\}_{k=1}^K$. Second, $\|\boldsymbol{U}_k^T \boldsymbol{x}_t\|$ is approximated by its expectation, i.e., $\mathbb{E}_{\boldsymbol{\epsilon}}[\|\boldsymbol{U}_k^T \boldsymbol{x}_t\|^2] = \mathbb{E}_{\boldsymbol{\epsilon}}\left[\|\boldsymbol{U}_k^T(s_t\boldsymbol{x}_0 + \gamma_t\boldsymbol{\epsilon})\|^2\right] = s_t^2\|\boldsymbol{U}_k^T\boldsymbol{x}_0\|^2 + \gamma_t^2 d$. We refer the reader to Appendix B.1 for more details on these approximation. Now, we are ready to show the following theorem.

**Theorem 3.** *Suppose that the DAE $\boldsymbol{x_\theta}(\cdot, t)$ in Problem (6) is parameterized into (16) for each $t \in [0, 1]$, where $\hat{w}_k(\boldsymbol{\theta}, \boldsymbol{x}_0)$ is defined in (17) for each $k \in [K]$. Then, Problem (6) is equivalent to the following subspace clustering problem:*

$$\max_{\boldsymbol{\theta}} \frac{1}{N} \sum_{k=1}^{K} \sum_{i \in C_k(\boldsymbol{\theta})} \|\boldsymbol{U}_k^T \boldsymbol{x}^{(i)}\|^2 \quad \text{s.t.} \quad [\boldsymbol{U}_1, \ldots, \boldsymbol{U}_K] \in \mathcal{O}^{n \times dK}, \tag{18}$$

*where $C_k(\boldsymbol{\theta}) := \{i \in [N] : \|\boldsymbol{U}_k^T \boldsymbol{x}^{(i)}\| \geq \|\boldsymbol{U}_l^T \boldsymbol{x}^{(i)}\|, \, \forall l \neq k\}$ for each $k \in [K]$.*

We defer the proof to Appendix B.2. When the DAE is parameterized into (16), Theorem 3 demonstrates that optimizing the training loss of diffusion models is equivalent to solving the subspace clustering problem (Vidal, 2011; Wang et al., 2022). Moreover, the equivalence allows us to characterize the required minimum number of samples for learning the underlying MoLRG distribution.

**Theorem 4.** *Consider the setting of Theorem 3. Suppose that the training samples $\{\boldsymbol{x}^{(i)}\}_{i=1}^{N}$ are generated by the MoLRG distribution in Definition 1. Suppose $d \gtrsim \log N$ and $\|\boldsymbol{e}_i\| \lesssim \sqrt{d/N}$ for all $i \in [N]$. Let $\{\hat{\boldsymbol{U}}_k\}_{k=1}^{K}$ denote an optimal solution of Problem (6) and $N_k$ denote the number of samples from the $k$-th Gaussian component. Then, the following statements hold:*

(i) *If $N_k \geq d$ for each $k \in [K]$, there exists a permutation $\Pi : [K] \to [K]$ such that with probability at least $1 - 2K^2 N^{-1} - \sum_{k=1}^{K} \left(1/2^{N_k - d + 1} + \exp\left(-c_2 N_k\right)\right)$ for each $k \in [K]$,*

$$\left\|\hat{\boldsymbol{U}}_{\Pi(k)} \hat{\boldsymbol{U}}_{\Pi(k)}^T - \boldsymbol{U}_k^\star \boldsymbol{U}_k^{\star T}\right\|_F \leq \frac{c_1 \sqrt{\sum_{i=1}^{N} \|\boldsymbol{e}_i\|^2}}{\sqrt{N_k} - \sqrt{d-1}}, \tag{19}$$

*where $c_1, c_2 > 0$ are constants that depend polynomially only on the Gaussian moment.*

(ii) *If $N_k < d$ for some $k \in [K]$, there exists a permutation $\Pi : [K] \to [K]$ and $k \in [K]$ such that with probability at least $1 - 2K^2 N^{-1} - \sum_{k=1}^{K} \left(1/2^{d - N_k + 1} + \exp\left(-c_2' N_k\right)\right)$,*

$$\left\|\hat{\boldsymbol{U}}_{\Pi(k)} \hat{\boldsymbol{U}}_{\Pi(k)}^T - \boldsymbol{U}_k^\star \boldsymbol{U}_k^{\star T}\right\|_F \geq \sqrt{2 \min\{d - N_k, n - d\}} - \frac{c_1' \sqrt{\sum_{i=1}^{N} \|\boldsymbol{e}_i\|^2}}{\sqrt{d} - \sqrt{N_k - 1}}, \tag{20}$$

*where $c_1', c_2' > 0$ are constants that depend polynomially only on the Gaussian*

**Remark 2.** We defer the proof to Appendix B.3. We discuss the implications of our results below.

- *Phase transition in learning the underlying distribution.* This theorem demonstrates that when the number of samples in each subspace exceeds the dimension of the subspace and the noise is bounded, the optimal solution of the training loss (6) under the parameterization (16) can recover the underlying subspaces up to the noise level. Conversely, when the number of samples is insufficient, there exists an optimal solution that may recover wrong subspaces; see Figures 4(c,d).

- *Connections to the phase transition from memorization to generalization.* We should clarify the difference between the phase transition described in Theorems 2 & 4 and the phase transition from memorization to generalization. Our phase transition refers to the shift from failure to success of learning the underlying distribution as the number of training samples increase, whereas the latter concerns the shift from memorizing data to generalizing from it as the number of training samples increases. Nevertheless, our theory still sheds light on the minimal number of samples required for diffusion models to enter the generalized regime.

## 4 Experiments & Practical Implications

In this section, we first investigate phase transitions of diffusion models in learning distributions under both theoretical and practical settings in Section 4.1. Next, we demonstrate the practical implications of our work by exploring the correspondence between low-dimensional subspaces and semantic representations for controllable image editing in Section 4.2.

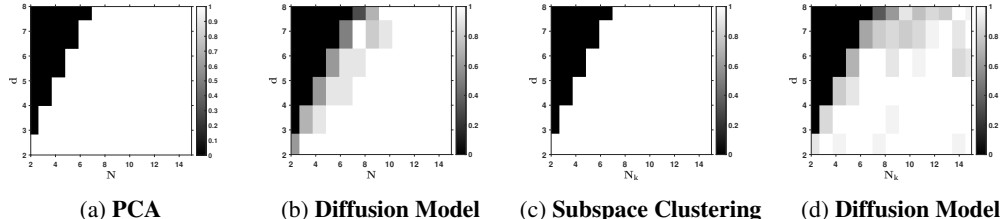

| (a) **PCA** | (b) **Diffusion Model** | (c) **Subspace Clustering** | (d) **Diffusion Model** |

Figure 4: **Phase transition of learning the MoLRG distribution.** The $x$-axis is the number of training samples and $y$-axis is the dimension of subspaces. Darker pixels represent a lower empirical probability of success. When $K = 1$, we apply SVD and train diffusion models to solve Problems (12) and (6), visualizing the results in (a) and (b), respectively. When $K = 2$, we apply a subspace clustering method and train diffusion models for solving Problems (18) and (6), visualizing the results in (c) and (d), respectively.

### 4.1 PHASE TRANSITION IN LEARNING DISTRIBUTIONS

In this subsection, we conduct experiments on both synthetic and real datasets to study the phase transition of diffusion models in learning distributions.

**Learning the MoLRG distribution with the theoretical parameterizations.** To begin, we optimize the training loss (6) with the theoretical parameterization (9), where the data samples are generated by the MoLRG distribution. First, we apply stochastic gradient descent (see Algorithm 1) to solve Problem (6) with the DAE parameterized as (9). For comparison, according to Theorem 1 (resp., Theorem 3), we apply a singular value decomposition (resp., subspace clustering (Wang et al., 2022)) to solve Problem (12) (resp, Problem (18)). We conduct three sets of experiments, where the data samples are respectively generated according to the single low-rank Gaussian distribution (10) with $K = 1$ and a mixture of low-rank Gaussian distributions (15) with $K = 2, 3$. In each set, we set the total dimension $n = 48$ and let the subspace dimension $d$ and the number of training samples $N$ vary from 2 to 8 and 2 to 15 with increments of 1, respectively. For every pair of $d$ and $N$, we generate 20 instances, run the above methods, and calculate the successful rate of recovering the underlying subspaces. The simulation results are visualized in Figure 4 and Figure 7. It is observed that all these methods exhibit a phase transition from failure to success in learning the subspaces as the number of training samples increases, which supports the results in Theorems 2 and 4.

**Learning the MoLRG distribution with U-Net.** Next, we optimize the training loss (6) with parameterizing the DAE $\boldsymbol{x}_{\boldsymbol{\theta}}(\cdot, t)$ using U-Net, detailed experiment settings are in Appendix D.2. We measure the generalization ability of U-Net via *generalization (GL) score* defined in Eq. (48). The trained diffusion model is in the memorization regime when the GL score is close to 0, while it is in the generalization regime when the GL score is close to 1. Detailed discussions about the metric are in Appendix D.2. In the experiments, we generate the data samples using the MoLRG distribution with $K = 2$, $n = 48$, and $d_k \in \{3, 4, 5, 6\}$. Then, we plot the GL score against the $N_k/d_k$ for each $d_k$ in Figure 5(a). It is observed that for a fixed $d_k$, the generalization performance of diffusion models improves as the number of training samples increases. Notably, for different values of $d_k$, the plot of the GL score against the $N_k/d_k$ remains approximately consistent. This observation indicates that the phase transition curve for U-Net learning the MoLRG distribution depends on the ratio $N_k/d_k$ rather than on $N_k$ and $d_k$ individually. When $N_k/d_k \approx 60$, GL score $\approx 1.0$ suggesting that U-Net generalizes when $N_k \geq 60d_k$. This linear relationship for the phase transition differs from $N_k \geq d_k$ in Theorem 4 due to training with U-Net instead of the optimal network parameterization in Eq. (9). Nevertheless, Theorem 2 and Theorem 4 still provide valuable insights into learning distributions via diffusion models by demonstrating a similar phase transition phenomenon and confirming a linear relationship between $N_k$ and $d_k$.

**Learning real image data distributions with U-Net.** Finally, we train diffusion models using U-Net on real image datasets AFHQ, CelebA, FFHQ, and CIFAR-10. The detailed experiment settings are deferred to Appendix D.3. we utilize the generalization (GL) score on the real-world image dataset according to Zhang et al. (2023). The definition of the metric is in Eq. (49) and detailed discussions are in Appendix D.3. Intuitively, GL score measures the dissimilarity between the generated sample $\boldsymbol{x}$ and all $N$ samples $\boldsymbol{y}_i$ from the training dataset $\{\boldsymbol{y}_i\}_{i=1}^N$. A higher GL score indicates stronger generalizability. For each data set, we train U-Net and plot the GL score against

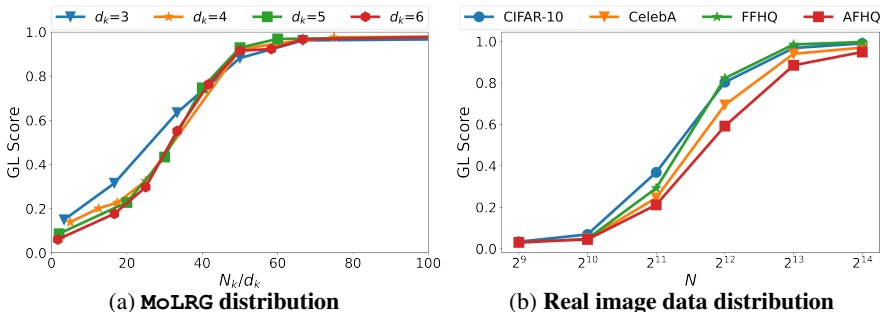

(a) `MoLRG` distribution    (b) **Real image data distribution**

Figure 5: **Phase transition of learning distributions via U-Net.** In (a), the $x$-axis is the number of training samples over the intrinsic dimension, while in (b), it is the total number of training samples. The $y$-axis is the GL score. We train diffusion models with the U-Net architecture on (a) the data samples generated by the `MoLRG` distribution with $K = 2$, $n = 48$ and $d_k$ varying from 3 to 6 and (b) real image datasets CIFAR-10, CelebA, FFHQ and AFHQ. The GL score is low when U-Net memorizes the training data and high when it learns the underlying distribution.

the number of training samples in Figure 5(b). The phase transition in the real dataset is illustrated in Figure 5(b). As observed, the order in which the samples need to generalize follows the relationship: AFHQ > CelebA > FFHQ $\approx$ CIFAR-10. Additionally, from our previous observations in Figure 3, the relationship of the intrinsic dimensions for these datasets is: AFHQ > FFHQ > CelebA $\approx$ CIFAR-10. Both AFHQ and CelebA align well with our theoretical analysis, which indicates that more samples are required for the model to generalize as the intrinsic dimension increases.

### 4.2 SEMANTIC MEANINGS OF LOW-DIMENSIONAL SUBSPACES

In this subsection, we conduct experiments to verify the correspondence between the low-dimensional subspaces of the data distribution and the semantics of images on real datasets. We denote the Jacobian of the DAE $x_{\theta}(x_t, t)$ by $J_t := \nabla_{x_t} x_{\theta}(x_t, t) \in \mathbb{R}^{n \times n}$ and let $J_t = U\Sigma V^T$ be an singular value decomposition (SVD) of $J_t$, where $r = \mathrm{rank}(J_t)$, $U = [u_1, \cdots, u_r] \in \mathcal{O}^{n \times r}$, $V = [v_1, \cdots, v_r] \in \mathcal{O}^{n \times r}$, and $\Sigma = \mathrm{diag}(\sigma_1, \ldots, \sigma_r)$ with $\sigma_1 \geq \cdots \geq \sigma_r$ being the singular values. To validate the semantic meaning of the basis vectors $v_i$, we vary the value of $\alpha$ from negative to positive and visualize the resulting changes in the generated images. In the experiments, we use a pre-trained diffusion denoising probabilistic model (DDPM) (Ho et al., 2020) on the Met-Faces dataset (Karras et al., 2020). We randomly select an image $x_0$ from this dataset and use the reverse process of the diffusion denoising implicit model (DDIM) (Song et al., 2020) to generate $x_t$ at $t = 0.7T$, where $T$ denote the total number of time steps. We respectively choose the changed direction as the leading right singular vectors $v_1, v_3, v_4, v_5, v_6$ and use $\tilde{x}_t = x_t + \alpha v_i$ to generate new images with $\alpha \in [-4, 4]$ shown in Figure 8. It is observed that these singular vectors enable different semantic edits in terms of gender, hairstyle, and color of the image. For comparison, we generate a random unit vector $s$ and move $x_t$ along the direction of $s$, where the editing strength $\alpha$ is the same as the semantic edits column-wise. The results are shown in the last column of Figure 2. Moving along random directions provides minimal semantic changes in the generated images, indicating that the low-dimensional subspace spanned by $V$ is non-trivial and corresponds to semantic meaningful image attributes. More experimental results can be found in Figure 8, Figure 9 in Appendix D.3.

## 5 CONCLUSION & DISCUSSION

In this work, we studied the training loss of diffusion models to investigate when and why diffusion models can learn the underlying distribution without suffering from the curse of dimensionality. Motivated by extensive empirical observations, we assumed that the underlying data distribution is a mixture of low-rank Gaussians. Specifically, we showed that minimizing the training loss is equivalent to solving the subspace clustering problem under proper network parameterization. Based on this equivalence, we further showed that the optimal solutions to the training loss can recover the underlying subspaces when the number of samples scales linearly with the intrinsic dimensionality of the data distribution. Moreover, we established the correspondence between the subspaces and semantic representations of image data. Since our studied network parameterization is not sufficiently over-parameterized, a future direction is to extend our analysis to an over-parameterized case to fully explain the transition from memorization to generalization.

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

# Supplementary Material

In the appendix, the organization is as follows. We first provide proof details for Section 2 and Section 3 in Appendix A and Appendix B, respectively. Then, we present our experimental setups for Figure 3 in Appendix C and for Section 4 in Appendix D. Finally, some auxiliary results for proving the main theorems are provided in Appendix E.

To simplify our development, we introduce some further notation. We denote by $\mathcal{N}(\boldsymbol{\mu}, \boldsymbol{\Sigma})$ a multivariate Gaussian distribution with mean $\boldsymbol{\mu} \in \mathbb{R}^n$ and covariance $\boldsymbol{\Sigma} \succeq \mathbf{0}$. Given a Gaussian random vector $\boldsymbol{x} \sim \mathcal{N}(\boldsymbol{\mu}, \boldsymbol{\Sigma})$, if $\boldsymbol{\Sigma} \succ \mathbf{0}$, with abuse of notation, we write its pdf as

$$\mathcal{N}(\boldsymbol{x}; \boldsymbol{\mu}, \boldsymbol{\Sigma}) := \frac{1}{(2\pi)^{n/2} \det^{1/2}(\boldsymbol{\Sigma})} \exp\left(-\frac{1}{2}(\boldsymbol{x} - \boldsymbol{\mu})^T \boldsymbol{\Sigma}^{-1}(\boldsymbol{x} - \boldsymbol{\mu})\right). \tag{21}$$

If a random vector $\boldsymbol{x} \in \mathbb{R}^n$ satisfies $\boldsymbol{x} \sim \mathcal{N}(\boldsymbol{\mu}, \boldsymbol{U}\boldsymbol{U}^T)$ for some $\boldsymbol{\mu} \in \mathbb{R}^n$ and $\boldsymbol{U} \in \mathcal{O}^{n \times d}$, we have

$$\boldsymbol{x} = \boldsymbol{\mu} + \boldsymbol{U}\boldsymbol{a}, \tag{22}$$

where $\boldsymbol{a} \sim \mathcal{N}(\mathbf{0}, \boldsymbol{I}_d)$. Therefore, a mixture of low-rank Gaussians in Definition 1 can be expressed as

$$\mathbb{P}\left(\boldsymbol{x} = \boldsymbol{U}_k^\star \boldsymbol{a}_k\right) = \pi_k, \text{ where } \boldsymbol{a}_k \sim \mathcal{N}(\mathbf{0}, \boldsymbol{I}_{d_k}), \ \forall k \in [K]. \tag{23}$$

## A PROOFS IN SECTION 2

### A.1 RELATION BETWEEN SCORE MATCHING LOSS AND DENOISER AUTOENCODER LOSS

To estimate $\nabla \log p_t(\boldsymbol{x})$, one can train a time-dependent score-based model $\boldsymbol{s}_{\boldsymbol{\theta}}(\boldsymbol{x}, t)$ via minimizing the following objective Song et al. (2021):

$$\min_{\boldsymbol{\theta}} \int_0^1 \xi_t \mathbb{E}_{\boldsymbol{x}_0 \sim p_{\text{data}}} \mathbb{E}_{\boldsymbol{x}_t | \boldsymbol{x}_0} \left[\|\boldsymbol{s}_{\boldsymbol{\theta}}(\boldsymbol{x}_t, t) - \nabla \log p_t(\boldsymbol{x}_t | \boldsymbol{x}_0)\|^2\right] \mathrm{d}t, \tag{24}$$

where $\xi_t : [0, 1] \to \mathbb{R}^+$ is a positive weighting function. Let $\boldsymbol{x}_{\boldsymbol{\theta}}(\cdot, t) : \mathbb{R}^d \times [0, 1] \to \mathbb{R}^d$ denote a neural network parameterized by parameters $\boldsymbol{\theta}$ to approximate $\mathbb{E}[\boldsymbol{x}_0 | \boldsymbol{x}_t]$. According to the Tweedie's formula (5), $\boldsymbol{s}_{\boldsymbol{\theta}}(\boldsymbol{x}_t, t) = (s_t \boldsymbol{x}_{\boldsymbol{\theta}}(\boldsymbol{x}_t, t) - \boldsymbol{x}_t)/\gamma_t^2$ can be used to estimate score functions. Substituting this and $\nabla \log p_t(\boldsymbol{x}_t | \boldsymbol{x}_0) = (s_t \boldsymbol{x}_0 - \boldsymbol{x}_t)/\gamma_t^2$ due to (3) yields

$$\min_{\boldsymbol{\theta}} \int_0^1 \xi_t \mathbb{E}_{\boldsymbol{x}_0 \sim p_{\text{data}}} \mathbb{E}_{\boldsymbol{x}_t | \boldsymbol{x}_0} \left[\left\|\frac{1}{\gamma_t^2}(s_t \boldsymbol{x}_{\boldsymbol{\theta}}(\boldsymbol{x}_t, t) - \boldsymbol{x}_t) - \frac{1}{\gamma_t^2}(s_t \boldsymbol{x}_0 - \boldsymbol{x}_t)\right\|^2\right] \mathrm{d}t$$

$$= \int_0^1 \frac{\xi_t}{s_t^2 \sigma_t^4} \mathbb{E}_{\boldsymbol{x}_0 \sim p_{\text{data}}} \mathbb{E}_{\boldsymbol{\epsilon} \sim \mathcal{N}(\mathbf{0}, \boldsymbol{I}_n)} \left[\|\boldsymbol{x}_{\boldsymbol{\theta}}(s_t \boldsymbol{x}_0 + \gamma_t \boldsymbol{\epsilon}, t) - \boldsymbol{x}_0\|^2\right] \mathrm{d}t,$$

where the equality follows from $\boldsymbol{x}_t = s_t \boldsymbol{x}_0 + \gamma_t \boldsymbol{\epsilon}$ due to (3). Then, we obtain

$$\min_{\boldsymbol{\theta}} \int_0^1 \lambda_t \mathbb{E}_{\boldsymbol{x}_0 \sim p_{\text{data}}} \mathbb{E}_{\boldsymbol{\epsilon} \sim \mathcal{N}(\mathbf{0}, \boldsymbol{I}_n)} \left[\|\boldsymbol{x}_{\boldsymbol{\theta}}(s_t \boldsymbol{x}_0 + \gamma_t \boldsymbol{\epsilon}, t) - \boldsymbol{x}_0\|^2\right] \mathrm{d}t, \tag{25}$$

where $\lambda_t = \xi_t/(s_t^2 \sigma_t^4)$. However, only data points $\{\boldsymbol{x}^{(i)}\}_{i=1}^N$ sampled from the underlying data distribution $p_{\text{data}}$ are available in practice. Therefore, we study the following empirical counterpart of Problem (25) over the training samples, i.e., Problem (6). We refer the reader to (Kadkhodaie et al., 2023, Section 2.1) for more discussions on the denoising error of this problem.

### A.2 PROOF OF IN LEMMA 1

Assuming that the underlying data distribution follows a mixture of low-rank Gaussians as defined in Definition 1, we first compute the ground-truth score function as follows.

**Proposition 1.** *Suppose that the underlying data distribution $p_{\text{data}}$ follows a mixture of low-rank Gaussian distributions in Definition 1. In the forward process of diffusion models, the pdf of $\boldsymbol{x}_t$ for each $t > 0$ is*

$$p_t(\boldsymbol{x}) = \sum_{k=1}^{K} \pi_k \mathcal{N}(\boldsymbol{x}; \boldsymbol{0}, s_t^2 \boldsymbol{U}_k^{\star} \boldsymbol{U}_k^{\star T} + \gamma_t^2 \boldsymbol{I}_n), \tag{26}$$

*where $\gamma_t = s_t \sigma_t$. Moreover, the score function of $p_t(\boldsymbol{x})$ is*

$$\nabla \log p_t(\boldsymbol{x}) = -\frac{1}{\gamma_t^2} \left( \boldsymbol{x} - \frac{s_t^2}{s_t^2 + \gamma_t^2} \frac{\sum_{k=1}^{K} \pi_k \mathcal{N}(\boldsymbol{x}; \boldsymbol{0}, s_t^2 \boldsymbol{U}_k^{\star} \boldsymbol{U}_k^{\star T} + \gamma_t^2 \boldsymbol{I}_n) \boldsymbol{U}_k^{\star} \boldsymbol{U}_k^{\star T} \boldsymbol{x}}{\sum_{k=1}^{K} \pi_k \mathcal{N}(\boldsymbol{x}; \boldsymbol{0}, s_t^2 \boldsymbol{U}_k^{\star} \boldsymbol{U}_k^{\star T} + \gamma_t^2 \boldsymbol{I}_n)} \right). \tag{27}$$

*Proof.* Let $Y \in \{1, \ldots, K\}$ be a discrete random variable that denotes the value of components of the mixture model. Note that $\gamma_t = s_t \sigma_t$. It follows from Definition 1 that $\mathbb{P}(Y = k) = \pi_k$ for each $k \in [K]$. We first compute

$$p_t(\boldsymbol{x}|Y = k) = \int p_t(\boldsymbol{x}|Y = k, \boldsymbol{a}_k) \mathcal{N}(\boldsymbol{a}_k; \boldsymbol{0}, \boldsymbol{I}_{d_k}) \, \mathrm{d}\boldsymbol{a}_k = \int p_t(\boldsymbol{x}|\boldsymbol{x}_0 = \boldsymbol{U}_k^{\star} \boldsymbol{a}_k) \mathcal{N}(\boldsymbol{a}_k; \boldsymbol{0}, \boldsymbol{I}_{d_k}) \, \mathrm{d}\boldsymbol{a}_k$$

$$= \int \mathcal{N}(\boldsymbol{x}; s_t \boldsymbol{U}_k^{\star} \boldsymbol{a}_k, \gamma_t^2 \boldsymbol{I}_n) \mathcal{N}(\boldsymbol{a}_k; \boldsymbol{0}, \boldsymbol{I}_{d_k}) \, \mathrm{d}\boldsymbol{a}_k$$

$$= \frac{1}{(2\pi)^{n/2}(2\pi)^{d_k/2}\gamma_t^n} \int \exp\left(-\frac{1}{2\gamma_t^2}\|\boldsymbol{x} - s_t \boldsymbol{U}_k^{\star} \boldsymbol{a}_k\|^2\right) \exp\left(-\frac{1}{2}\|\boldsymbol{a}_k\|^2\right) \mathrm{d}\boldsymbol{a}_k$$

$$= \frac{1}{(2\pi)^{n/2}\gamma_t^n} \left(\frac{s_t^2 + \gamma_t^2}{\gamma_t^2}\right)^{-d/2} \exp\left(-\frac{1}{2\gamma_t^2} \boldsymbol{x}^T \left(\boldsymbol{I}_n - \frac{s_t^2}{s_t^2 + \gamma_t^2} \boldsymbol{U}_k^{\star} \boldsymbol{U}_k^{\star T}\right) \boldsymbol{x}\right)$$

$$\int \frac{1}{(2\pi)^{d_k/2}} \left(\frac{\gamma_t^2}{s_t^2 + \gamma_t^2}\right)^{-d/2} \exp\left(-\frac{s_t^2 + \gamma_t^2}{2\gamma_t^2} \left\|\boldsymbol{a}_k - \frac{s_t}{s_t^2 + \gamma_t^2} \boldsymbol{U}_k^{\star T} \boldsymbol{x}\right\|^2\right) \mathrm{d}\boldsymbol{a}_k$$

$$= \frac{1}{(2\pi)^{n/2}} \frac{1}{\left((s_t^2 + \gamma_t^2)^d \gamma_t^{2(n-d)}\right)^{1/2}} \exp\left(-\frac{1}{2\gamma_t^2} \boldsymbol{x}^T \left(\boldsymbol{I}_n - \frac{s_t^2}{s_t^2 + \gamma_t^2} \boldsymbol{U}^{\star} \boldsymbol{U}^{\star T}\right) \boldsymbol{x}\right)$$

$$= \frac{1}{(2\pi)^{n/2} \det^{1/2}(s_t^2 \boldsymbol{U}_k^{\star} \boldsymbol{U}_k^{\star T} + \gamma_t^2 \boldsymbol{I}_n)} \exp\left(-\frac{1}{2} \boldsymbol{x}^T \left(s_t^2 \boldsymbol{U}_k^{\star} \boldsymbol{U}_k^{\star T} + \gamma_t^2 \boldsymbol{I}_n\right)^{-1} \boldsymbol{x}\right)$$

$$= \mathcal{N}(\boldsymbol{x}; \boldsymbol{0}, s_t^2 \boldsymbol{U}_k^{\star} \boldsymbol{U}_k^{\star T} + \gamma_t^2 \boldsymbol{I}_n),$$

where the second equality follows from (3), the third equality uses (21), the fourth equality is due to the fact that $\langle \boldsymbol{x}, \boldsymbol{U}_k^{\star} \boldsymbol{a} \rangle$ is an odd function, and the second to last equality uses $\det(s_t^2 \boldsymbol{U}_k^{\star} \boldsymbol{U}_k^{\star T} + \gamma_t^2 \boldsymbol{I}_n) = (s_t^2 + \gamma_t^2)^d \gamma_t^{2(n-d)}$ and $(s_t^2 \boldsymbol{U}_k^{\star} \boldsymbol{U}_k^{\star T} + \gamma_t^2 \boldsymbol{I}_n)^{-1} = \left(\boldsymbol{I}_n - s_t^2/(s_t^2 + \gamma_t^2) \boldsymbol{U}_k^{\star} \boldsymbol{U}_k^{\star T}\right)/\gamma_t^2$ due to the matrix inversion lemma and $\boldsymbol{U}_k^{\star T} \boldsymbol{U}_k^{\star} = \boldsymbol{I}_{d_k}$. This, together with $\mathbb{P}(Y = k) = \pi_k$ for each $k \in [K]$, yields

$$p_t(\boldsymbol{x}) = \sum_{k=1}^{K} p_t(\boldsymbol{x}|Y = k) \mathbb{P}(Y = k) = \sum_{k=1}^{K} \pi_k \mathcal{N}(\boldsymbol{x}; \boldsymbol{0}, s_t^2 \boldsymbol{U}_k^{\star} \boldsymbol{U}_k^{\star T} + \gamma_t^2 \boldsymbol{I}_n).$$

Next, we directly compute

$$\nabla \log p_t(\boldsymbol{x}) = \frac{\nabla p_t(\boldsymbol{x})}{p_t(\boldsymbol{x})} = \frac{\sum_{k=1}^{K} \pi_k \mathcal{N}(\boldsymbol{x}; \boldsymbol{0}, s_t^2 \boldsymbol{U}_k^{\star} \boldsymbol{U}_k^{\star T} + \gamma_t^2 \boldsymbol{I}_n) \left(-\frac{1}{\gamma_t^2} \boldsymbol{x} + \frac{s_t^2}{\gamma_t^2(s_t^2 + \gamma_t^2)} \boldsymbol{U}_k^{\star} \boldsymbol{U}_k^{\star T} \boldsymbol{x}\right)}{\sum_{k=1}^{K} \pi_k \mathcal{N}(\boldsymbol{x}; \boldsymbol{0}, s_t^2 \boldsymbol{U}_k^{\star} \boldsymbol{U}_k^{\star T} + \gamma_t^2 \boldsymbol{I}_n)}$$

$$= -\frac{1}{\gamma_t^2} \left( \boldsymbol{x} - \frac{s_t^2}{s_t^2 + \gamma_t^2} \frac{\sum_{k=1}^{K} \pi_k \mathcal{N}(\boldsymbol{x}; \boldsymbol{0}, s_t^2 \boldsymbol{U}_k^{\star} \boldsymbol{U}_k^{\star T} + \gamma_t^2 \boldsymbol{I}_n) \boldsymbol{U}_k^{\star} \boldsymbol{U}_k^{\star T} \boldsymbol{x})}{\sum_{k=1}^{K} \pi_k \mathcal{N}(\boldsymbol{x}; \boldsymbol{0}, s_t^2 \boldsymbol{U}_k^{\star} \boldsymbol{U}_k^{\star T} + \gamma_t^2 \boldsymbol{I}_n)} \right).$$

$\square$

*Proof of Lemma 1.* According to (5) and Proposition 1, we compute

$$\mathbb{E}\left[\boldsymbol{x}_0|\boldsymbol{x}_t\right] = \frac{\boldsymbol{x}_t + \gamma_t^2 \nabla \log p_t(\boldsymbol{x}_t)}{s_t} = \frac{s_t}{s_t^2 + \gamma_t^2} \frac{\sum_{k=1}^K \pi_k \mathcal{N}(\boldsymbol{x}; \boldsymbol{0}, s_t^2 \boldsymbol{U}_k^\star \boldsymbol{U}_k^{\star T} + \gamma_t^2 \boldsymbol{I}_n) \boldsymbol{U}_k^\star \boldsymbol{U}_k^{\star T} \boldsymbol{x}_t}{\sum_{k=1}^K \pi_k \mathcal{N}(\boldsymbol{x}_t; \boldsymbol{0}, s_t^2 \boldsymbol{U}_k^\star \boldsymbol{U}_k^{\star T} + \gamma_t^2 \boldsymbol{I}_n)}$$

$$= \frac{s_t}{s_t^2 + \gamma_t^2} \frac{\sum_{k=1}^K \pi_k \exp\left(-\frac{1}{2\gamma_t^2}\left(\|\boldsymbol{x}_t\|^2 - \frac{s_t^2}{s_t^2 + \gamma_t^2}\|\boldsymbol{U}_k^{\star T}\boldsymbol{x}_t\|^2\right)\right) \boldsymbol{U}_k^\star \boldsymbol{U}_k^{\star T} \boldsymbol{x}_t}{\sum_{k=1}^K \pi_k \exp\left(-\frac{1}{2\gamma_t^2}\left(\|\boldsymbol{x}_t\|^2 - \frac{s_t^2}{s_t^2 + \gamma_t^2}\|\boldsymbol{U}_k^{\star T}\boldsymbol{x}_t\|^2\right)\right)}$$

$$= \frac{s_t}{s_t^2 + \gamma_t^2} \frac{\sum_{k=1}^K \pi_k \exp\left(\frac{1}{2\gamma_t^2}\frac{s_t^2}{s_t^2 + \gamma_t^2}\|\boldsymbol{U}_k^{\star T}\boldsymbol{x}_t\|^2\right) \boldsymbol{U}_k^\star \boldsymbol{U}_k^{\star T} \boldsymbol{x}_t}{\sum_{k=1}^K \pi_k \exp\left(\frac{1}{2\gamma_t^2}\frac{s_t^2}{s_t^2 + \gamma_t^2}\|\boldsymbol{U}_k^{\star T}\boldsymbol{x}_t\|^2\right)},$$

where the third equality uses (21) and $\left(s_t^2 \boldsymbol{U}_k^\star \boldsymbol{U}_k^{\star T} + \gamma_t^2 \boldsymbol{I}_n\right)^{-1} = \left(\boldsymbol{I}_n - s_t^2/(s_t^2 + \gamma_t^2)\boldsymbol{U}_k^\star \boldsymbol{U}_k^{\star T}\right)/\gamma_t^2$ due to the matrix inversion lemma. □

## A.3 PROOF OF THEOREM 1

*Proof of Theorem 1.* Plugging (11) into the integrand of (6) yields

$$\mathbb{E}_{\boldsymbol{\epsilon}}\left[\left\|\frac{s_t}{s_t^2 + \gamma_t^2}\boldsymbol{U}\boldsymbol{U}^T\left(s_t \boldsymbol{x}^{(i)} + \gamma_t \boldsymbol{\epsilon}\right) - \boldsymbol{x}^{(i)}\right\|^2\right]$$

$$= \left\|\frac{s_t^2}{s_t^2 + \gamma_t^2}\boldsymbol{U}\boldsymbol{U}^T\boldsymbol{x}^{(i)} - \boldsymbol{x}^{(i)}\right\|^2 + \frac{(s_t\gamma_t)^2}{(s_t^2 + \gamma_t)^2}\mathbb{E}_{\boldsymbol{\epsilon}}\left[\|\boldsymbol{U}\boldsymbol{U}^T\boldsymbol{\epsilon}\|^2\right]$$

$$= \left\|\frac{s_t^2}{s_t^2 + \gamma_t^2}\boldsymbol{U}\boldsymbol{U}^T\boldsymbol{x}^{(i)} - \boldsymbol{x}^{(i)}\right\|^2 + \frac{(s_t\gamma_t)^2 d}{(s_t^2 + \gamma_t)^2},$$

where the first equality follows from $\mathbb{E}_{\boldsymbol{\epsilon}}[\langle \boldsymbol{x}, \boldsymbol{\epsilon}\rangle] = 0$ for any given $\boldsymbol{x} \in \mathbb{R}^n$ due to $\boldsymbol{\epsilon} \sim \mathcal{N}(\boldsymbol{0}, \boldsymbol{I}_n)$, and the second equality uses $\mathbb{E}_{\boldsymbol{\epsilon}}\left[\|\boldsymbol{U}\boldsymbol{U}^T\boldsymbol{\epsilon}\|^2\right] = \mathbb{E}_{\boldsymbol{\epsilon}}\left[\|\boldsymbol{U}^T\boldsymbol{\epsilon}\|^2\right] = \sum_{i=1}^d \mathbb{E}_{\boldsymbol{\epsilon}}\left[\|\boldsymbol{u}_i^T\boldsymbol{\epsilon}\|^2\right] = d$ due to $\boldsymbol{U} \in \mathcal{O}^{n \times d}$ and $\boldsymbol{\epsilon} \sim \mathcal{N}(\boldsymbol{0}, \boldsymbol{I}_n)$. This, together with $\gamma_t = s_t\sigma_t$ and (6), yields

$$\ell(\boldsymbol{U}) = \frac{1}{N}\sum_{i=1}^N \int_0^1 \lambda_t \left(\|\boldsymbol{x}^{(i)}\|^2 - \frac{1 + 2\sigma_t^2}{(1 + \sigma_t^2)^2}\|\boldsymbol{U}^T\boldsymbol{x}^{(i)}\|^2 + \frac{\sigma_t^2 d}{(1 + \sigma_t^2)^2}\right) \mathrm{d}t,$$

Obviously, minimizing the above function in terms of $\boldsymbol{U}$ amounts to

$$\min_{\boldsymbol{U}^T\boldsymbol{U}=\boldsymbol{I}_d} -\int_0^1 \frac{(1 + 2\sigma_t^2)\lambda_t}{(1 + \sigma_t^2)^2}\mathrm{d}t \frac{1}{N}\sum_{i=1}^N \|\boldsymbol{U}^T\boldsymbol{x}^{(i)}\|^2,$$

which is equivalent to Problem (12). □

## A.4 PROOF OF THEOREM 2

*Proof of Theorem 2.* For ease of exposition, let

$$\boldsymbol{X} = \begin{bmatrix} \boldsymbol{x}^{(1)} & \dots & \boldsymbol{x}^{(N)} \end{bmatrix} \in \mathbb{R}^{n \times N},\ \boldsymbol{A} = \begin{bmatrix} \boldsymbol{a}_1 & \dots & \boldsymbol{a}_N \end{bmatrix} \in \mathbb{R}^{d \times N},\ \boldsymbol{E} = \begin{bmatrix} \boldsymbol{e}_1 & \dots & \boldsymbol{e}_N \end{bmatrix} \in \mathbb{R}^{n \times N}.$$

Using this and (10), we obtain

$$\boldsymbol{X} = \boldsymbol{U}^\star \boldsymbol{A} + \boldsymbol{E}. \tag{28}$$

Let $r_A := \text{rank}(\boldsymbol{A}) \le \min\{d, N\}$ and $\boldsymbol{A} = \boldsymbol{U}_A \boldsymbol{\Sigma}_A \boldsymbol{V}_A^T$ be an singular value decomposition (SVD) of $\boldsymbol{A}$, where $\boldsymbol{U}_A \in \mathcal{O}^{d \times r_A}$, $\boldsymbol{V}_A \in \mathcal{O}^{N \times r_A}$, and $\boldsymbol{\Sigma}_A \in \mathbb{R}^{r_A \times r_A}$. It follows from Theorem 1 that Problem (6) with the parameterization (11) is equivalent to Problem (12).

(i) Suppose that $N \geq d$. Applying Lemma 3 with $\varepsilon = 1/(2c_1)$ to $\boldsymbol{A} \in \mathbb{R}^{d \times N}$, it holds with probability at least $1 - 1/2^{N-d+1} - \exp(-c_2 N)$ that

$$\sigma_{\min}(\boldsymbol{A}) = \sigma_d(\boldsymbol{A}) \geq \frac{\sqrt{N} - \sqrt{d-1}}{2c_1}, \tag{29}$$

where $c_1, c_2 > 0$ are constants depending polynomially only on the Gaussian moment. This implies $r_A = d$ and $\boldsymbol{U}_A \in \mathcal{O}^d$. Since Problem (12) is a PCA problem, the columns of any optimal solution $\hat{\boldsymbol{U}} \in \mathcal{O}^{n \times d}$ consist of left singular vectors associated with the top $d$ singular values of $\boldsymbol{X}$. This, together with Wedin's Theorem (Wedin, 1972) and (28), yields

$$\left\| \hat{\boldsymbol{U}}\hat{\boldsymbol{U}}^T - \boldsymbol{U}^\star \boldsymbol{U}^{\star T} \right\|_F = \left\| \hat{\boldsymbol{U}}\hat{\boldsymbol{U}}^T - (\boldsymbol{U}^\star \boldsymbol{U}_A)(\boldsymbol{U}^\star \boldsymbol{U}_A)^T \right\|_F \leq \frac{2\|\boldsymbol{E}\|_F}{\sigma_{\min}(\boldsymbol{A})} = \frac{4c_1\|\boldsymbol{E}\|_F}{\sqrt{N} - \sqrt{d-1}}.$$

This, together with absorbing 4 into $c_1$, yields (13).

(ii) Suppose that $N < d$. According to Lemma 3 with $\varepsilon = 1/(2c_1)$, it holds with probability at least $1 - 1/2^{d-N+1} - \exp(-c_2 d)$ that

$$\sigma_{\min}(\boldsymbol{A}) = \sigma_N(\boldsymbol{A}) \geq \frac{\sqrt{d} - \sqrt{N-1}}{2c_1}, \tag{30}$$

where $c_1, c_2 > 0$ are constants depending polynomially only on the Gaussian moment. This implies $r_A = N$ and $\boldsymbol{U}_A \in \mathcal{O}^{d \times N}$. This, together with the fact that $\boldsymbol{A} = \boldsymbol{U}_A \boldsymbol{\Sigma}_A \boldsymbol{V}_A^T$ is an SVD of $\boldsymbol{A}$, yields that $\boldsymbol{U}^\star \boldsymbol{A} = (\boldsymbol{U}^\star \boldsymbol{U}_A)\boldsymbol{\Sigma}_A \boldsymbol{V}_A^T$ is an SVD of $\boldsymbol{U}^\star \boldsymbol{A}$ with $\boldsymbol{U}^\star \boldsymbol{U}_A \in \boldsymbol{O}^{n \times N}$. Note that $\text{rank}(\boldsymbol{X}) \leq N$. Let $\boldsymbol{X} = \boldsymbol{U}_X \boldsymbol{\Sigma}_X \boldsymbol{V}_X^T$ be an SVD of $\boldsymbol{X}$, where $\boldsymbol{U}_X \in \mathcal{O}^{n \times N}$, $\boldsymbol{V}_X \in \mathcal{O}^N$, and $\boldsymbol{\Sigma}_X \in \mathbb{R}^{N \times N}$. This, together with Wedin's Theorem (Wedin, 1972) and (30), yields

$$\left\| \boldsymbol{U}_X \boldsymbol{U}_X^T - \boldsymbol{U}^\star \boldsymbol{U}_A \boldsymbol{U}_A^T \boldsymbol{U}^{\star T} \right\|_F \leq \frac{2\|\boldsymbol{E}\|_F}{\sigma_{\min}(\boldsymbol{A})} = \frac{4c_1\|\boldsymbol{E}\|_F}{\sqrt{d} - \sqrt{N-1}}. \tag{31}$$

Note that Problem (12) has infinite optimal solutions when $N < d$, which take the form of

$$\hat{\boldsymbol{U}} = \begin{bmatrix} \boldsymbol{U}_X & \bar{\boldsymbol{U}}_X \end{bmatrix} \in \mathcal{O}^{n \times d}.$$

Now, we consider that $\bar{\boldsymbol{U}}_X \in \mathcal{O}^{n \times (d-N)}$ is an optimal solution of the following problem:

$$\min_{\boldsymbol{V} \in \mathcal{O}^{n \times (d-N)}, \boldsymbol{U}_X^T \boldsymbol{V} = \boldsymbol{0}} \|\boldsymbol{V}^T \boldsymbol{U}^\star (\boldsymbol{I} - \boldsymbol{U}_A \boldsymbol{U}_A^T)\|_F^2. \tag{32}$$

Then, one can verify that the rank of the following matrix is at most $d$:

$$\boldsymbol{B} := \begin{bmatrix} \boldsymbol{U}_X & \boldsymbol{U}^\star (\boldsymbol{I} - \boldsymbol{U}_A \boldsymbol{U}_A^T) \end{bmatrix}$$

Then, if $n \geq 2d - N$, it is easy to see that the optimal value of Problem (32) is 0. If $n < 2d - N$, the optima value is achieved at $\boldsymbol{V}^\star = [\boldsymbol{V}_1^\star \ \boldsymbol{V}_2^\star]$ with $\boldsymbol{V}_1^\star \in \mathbb{R}^{n \times (n-d)}$ and $\boldsymbol{V}_2^\star \in \mathbb{R}^{n \times (2d-N-n)}$ satisfying $\boldsymbol{V}_1^{\star T} \boldsymbol{B} = \boldsymbol{0}$, which implies

$$\|\boldsymbol{V}^{\star T} \boldsymbol{U}^\star (\boldsymbol{I} - \boldsymbol{U}_A \boldsymbol{U}_A^T)\|_F^2 = \|\boldsymbol{V}_2^{\star T} \boldsymbol{U}^\star (\boldsymbol{I} - \boldsymbol{U}_A \boldsymbol{U}_A^T)\|_F^2 \leq 2d - N - n.$$

Consequently, the optimal value of Problem (32) is less than

$$\max\{0, 2d - (n+N)\} \tag{33}$$

Then, we obtain that

$$\begin{aligned}
\left\| \hat{\boldsymbol{U}}\hat{\boldsymbol{U}}^T - \boldsymbol{U}^\star \boldsymbol{U}^{\star T} \right\|_F &= \|\boldsymbol{U}_X \boldsymbol{U}_X^T + \bar{\boldsymbol{U}}_X \bar{\boldsymbol{U}}_X^T - \boldsymbol{U}^\star \boldsymbol{U}_A \boldsymbol{U}_A^T \boldsymbol{U}^{\star T} - \boldsymbol{U}^\star (\boldsymbol{I} - \boldsymbol{U}_A \boldsymbol{U}_A^T)\boldsymbol{U}^{\star T}\| \\
&\geq \|\bar{\boldsymbol{U}}_X \bar{\boldsymbol{U}}_X^T - \boldsymbol{U}^\star (\boldsymbol{I} - \boldsymbol{U}_A \boldsymbol{U}_A^T)\boldsymbol{U}^{\star T}\|_F - \|\boldsymbol{U}_X \boldsymbol{U}_X^T - \boldsymbol{U}^\star \boldsymbol{U}_A \boldsymbol{U}_A^T \boldsymbol{U}^{\star T}\|_F \\
&\geq \sqrt{2(d-N) - 2\max\{0, 2d - (n+N)\}} - \frac{4c_1\|\boldsymbol{E}\|_F}{\sqrt{d} - \sqrt{N-1}} \\
&\geq \sqrt{2\min\{d-N, n-d\}} - \frac{4c_1\|\boldsymbol{E}\|_F}{\sqrt{d} - \sqrt{N-1}},
\end{aligned}$$

where the second inequality follows from $\bar{\boldsymbol{U}}_X = \boldsymbol{V}^\star$ and (33). Then, we complete the proof.

$\square$

## B   PROOFS IN SECTION 3.2

### B.1   THEORETICAL JUSTIFICATION OF THE DAE (16)

Since $\boldsymbol{x}_t = s_t \boldsymbol{x}_0 + \gamma_t \boldsymbol{\epsilon}$, we compute

$$\mathbb{E}_{\boldsymbol{\epsilon}} \left[ \| \boldsymbol{U}_k^T (s_t \boldsymbol{x}_0 + \gamma_t \boldsymbol{\epsilon}) \|^2 \right] = s_t^2 \| \boldsymbol{U}_k^T \boldsymbol{x}_0 \|^2 + \gamma_t^2 \mathbb{E}_{\boldsymbol{\epsilon}} [\| \boldsymbol{U}_k^T \boldsymbol{\epsilon} \|^2] = s_t^2 \| \boldsymbol{U}_k^T \boldsymbol{x}_0 \|^2 + \gamma_t^2 d,$$

where the first equality is due to $\boldsymbol{\epsilon} \sim \mathcal{N}(\boldsymbol{0}, \boldsymbol{I}_n)$ and $\mathbb{E}_{\boldsymbol{\epsilon}}[\langle \boldsymbol{U}_k^T \boldsymbol{x}_0, \boldsymbol{U}_k^T \boldsymbol{\epsilon} \rangle] = \boldsymbol{0}$ for each $k \in [K]$. This implies that when $n$ is sufficiently large, we can approximate $w_k(\boldsymbol{\theta}; \boldsymbol{x}_t)$ in (9) well by

$$w_k(\boldsymbol{\theta}; \boldsymbol{x}_t) \approx \frac{\exp \left( \phi_t \left( s_t^2 \| \boldsymbol{U}_k^T \boldsymbol{x}_0 \|^2 + \gamma_t^2 d \right) \right)}{\sum_{l=1}^K \exp \left( \phi_t \left( s_t^2 \| \boldsymbol{U}_l^T \boldsymbol{x}_0 \|^2 + \gamma_t^2 d \right) \right)}.$$

This soft-max function can be further approximated by the hard-max function. Therefore, we directly obtain (17).

### B.2   PROOF OF THEOREM 3

Equipped with the above setup, we are ready to prove Theorem 3.

*Proof of Theorem 3.* Plugging (16) into the integrand of (6) yields

$$\mathbb{E}_{\boldsymbol{\epsilon}} \left[ \left\| \frac{s_t}{s_t^2 + \gamma_t^2} \sum_{k=1}^K \hat{w}_k(\boldsymbol{\theta}; \boldsymbol{x}^{(i)}) \boldsymbol{U}_k \boldsymbol{U}_k^T (s_t \boldsymbol{x}^{(i)} + \gamma_t \boldsymbol{\epsilon}) - \boldsymbol{x}^{(i)} \right\|^2 \right]$$

$$= \left\| \frac{s_t^2}{s_t^2 + \gamma_t^2} \sum_{k=1}^K \hat{w}_k(\boldsymbol{\theta}; \boldsymbol{x}^{(i)}) \boldsymbol{U}_k \boldsymbol{U}_k^T \boldsymbol{x}^{(i)} - \boldsymbol{x}^{(i)} \right\|^2 + \frac{(s_t \gamma_t)^2}{(s_t^2 + \gamma_t^2)^2} \mathbb{E}_{\boldsymbol{\epsilon}} \left[ \left\| \sum_{k=1}^K \hat{w}_k(\boldsymbol{\theta}; \boldsymbol{x}^{(i)}) \boldsymbol{U}_k \boldsymbol{U}_k^T \boldsymbol{\epsilon} \right\|^2 \right]$$

$$= \frac{s_t^2}{s_t^2 + \gamma_t^2} \sum_{k=1}^K \left( \frac{s_t^2}{s_t^2 + \gamma_t^2} \hat{w}_k^2(\boldsymbol{\theta}; \boldsymbol{x}^{(i)}) - 2\hat{w}_k(\boldsymbol{\theta}; \boldsymbol{x}^{(i)}) \right) \| \boldsymbol{U}_k^T \boldsymbol{x}^{(i)} \|^2 + \| \boldsymbol{x}^{(i)} \|^2 + \frac{(s_t \gamma_t)^2 d}{(s_t^2 + \gamma_t^2)^2} \sum_{k=1}^K \hat{w}_k(\boldsymbol{\theta}; \boldsymbol{x}^{(i)}),$$

where the first equality follows from $\mathbb{E}_{\boldsymbol{\epsilon}}[\langle \boldsymbol{x}, \boldsymbol{\epsilon} \rangle] = 0$ for any fixed $\boldsymbol{x} \in \mathbb{R}^n$ due to $\boldsymbol{\epsilon} \sim \mathcal{N}(\boldsymbol{0}, \boldsymbol{I}_n)$, and the last equality uses $\boldsymbol{U}_k \in \mathcal{O}^{n \times d}$ and $\boldsymbol{U}_k^T \boldsymbol{U}_l = \boldsymbol{0}$ for all $k \neq l$. This, together with (6) and $\gamma_t = s_t \sigma_t$, yields

$$\ell(\boldsymbol{\theta}) = \frac{1}{N} \sum_{i=1}^N \sum_{k=1}^K \int_0^1 \frac{\lambda_t}{1 + \sigma_t^2} \left( \frac{1}{1 + \sigma_t^2} \hat{w}_k^2(\boldsymbol{\theta}; \boldsymbol{x}^{(i)}) - 2\hat{w}_k(\boldsymbol{\theta}; \boldsymbol{x}^{(i)}) \right) \mathrm{d}t \| \boldsymbol{U}_k^T \boldsymbol{x}^{(i)} \|^2 +$$

$$\frac{1}{N} \int_0^1 \lambda_t \mathrm{d}t \sum_{i=1}^N \| \boldsymbol{x}^{(i)} \|^2 + \left( \int_0^1 \frac{\sigma_t^2 \lambda_t}{(1 + \sigma_t^2)^2} \mathrm{d}t \right) \frac{d}{N} \sum_{i=1}^N \sum_{k=1}^K \hat{w}_k^2(\boldsymbol{\theta}; \boldsymbol{x}^{(i)}).$$

According to (16), we can partition $[N]$ into $\{C_k(\boldsymbol{\theta})\}_{k=1}^K$, where $C_k(\boldsymbol{\theta})$ for each $k \in [K]$ is defined as follows:

$$C_k(\boldsymbol{\theta}) := \left\{ i \in [N] : \| \boldsymbol{U}_k^T \boldsymbol{x}^{(i)} \| \geq \| \boldsymbol{U}_l^T \boldsymbol{x}^{(i)} \|, \ \forall l \neq k \right\}, \forall k \in [K]. \tag{34}$$

Then, we obtain

$$\sum_{i=1}^N \sum_{k=1}^K \hat{w}_k^2(\boldsymbol{\theta}; \boldsymbol{x}^{(i)}) = \sum_{k=1}^K \sum_{i \in C_k(\boldsymbol{\theta})} 1 = N.$$

This, together with plugging (34) into the above loss function, yields minimizing $\ell(\boldsymbol{\theta})$ is equivalent to minimizing

$$\frac{1}{N} \sum_{i=1}^N \sum_{k=1}^K \int_0^1 \frac{\lambda_t}{1 + \sigma_t^2} \left( \frac{1}{1 + \sigma_t^2} \hat{w}_k^2(\boldsymbol{\theta}; \boldsymbol{x}^{(i)}) - 2\hat{w}_k(\boldsymbol{\theta}; \boldsymbol{x}^{(i)}) \right) \mathrm{d}t \| \boldsymbol{U}_k^T \boldsymbol{x}^{(i)} \|^2$$

$$= \left( \int_0^1 \frac{\lambda_t}{1 + \sigma_t^2} \left( \frac{1}{1 + \sigma_t^2} - 2 \right) \mathrm{d}t \right) \frac{1}{N} \sum_{k=1}^K \sum_{i \in C_k(\boldsymbol{\theta})} \| \boldsymbol{U}_k^T \boldsymbol{x}^{(i)} \|^2.$$

Since $\frac{\lambda_t}{1+\sigma_t^2}\left(\frac{1}{1+\sigma_t^2}-2\right)<0$ for all $t\in[0,1]$, minimizing the above function is equivalent to

$$\max_{\boldsymbol{\theta}}\frac{1}{N}\sum_{k=1}^K\sum_{i\in C_k(\boldsymbol{\theta})}\|\boldsymbol{U}_k^T\boldsymbol{x}^{(i)}\|^2\qquad\text{s.t. }[\boldsymbol{U}_1\ \ldots\ \boldsymbol{U}_K]\in\mathcal{O}^{n\times dK}.$$

Then, we complete the proof. $\qquad\square$

### B.3 Proof of Theorem 4

*Proof of Theorem 4.* For ease of exposition, let $\delta:=\max\{\|\boldsymbol{e}_i\|:i\in[N]\}$,

$$f(\boldsymbol{\theta}):=\sum_{k=1}^K\sum_{i\in C_k(\boldsymbol{\theta})}\|\boldsymbol{U}_k^T\boldsymbol{x}^{(i)}\|^2,$$

and for each $k\in[K]$,

$$C_k^\star:=\left\{i\in[N]:\boldsymbol{x}^{(i)}=\boldsymbol{U}_k^\star\boldsymbol{a}_i+\boldsymbol{e}_i\right\}.$$

Suppose that (51) and (52) hold with $\boldsymbol{V}=\hat{\boldsymbol{U}}_k$ for all $i\in[N]$ and $k\neq l\in[K]$, which happens with probability $1-2K^2N^{-1}$ according to Lemma 5. This implies that for all $i\in[N]$ and $k\neq l\in[K]$,

$$\sqrt{d}-(2\sqrt{\log N}+2)\leq\|\boldsymbol{a}_i\|\leq\sqrt{d}+(2\sqrt{\log N}+2),\tag{35}$$

$$\|\hat{\boldsymbol{U}}_k^T\boldsymbol{U}_l^\star\|_F-(2\sqrt{\log N}+2)\leq\|\hat{\boldsymbol{U}}_k^T\boldsymbol{U}_l^\star\boldsymbol{a}_i\|\leq\|\hat{\boldsymbol{U}}_k^T\boldsymbol{U}_l^\star\|_F+(2\sqrt{\log N}+2).\tag{36}$$

Recall that the underlying basis matrices are denoted by $\boldsymbol{\theta}^\star=\{\boldsymbol{U}_k^\star\}_{k=1}^K$ and the optimal basis matrices are denoted by $\hat{\boldsymbol{\theta}}=\{\hat{\boldsymbol{U}}_k\}_{k=1}^K$.

First, we claim that $C_k(\boldsymbol{\theta}^\star)=C_k^\star$ for each $k\in[K]$. Indeed, for each $i\in C_k^\star$, we compute

$$\|\boldsymbol{U}_k^{\star T}\boldsymbol{x}^{(i)}\|=\|\boldsymbol{U}_k^{\star T}(\boldsymbol{U}_k^\star\boldsymbol{a}_i+\boldsymbol{e}_i)\|=\|\boldsymbol{a}_i+\boldsymbol{U}_k^{\star T}\boldsymbol{e}_i\|\geq\|\boldsymbol{a}_i\|-\|\boldsymbol{e}_i\|,\tag{37}$$

$$\|\boldsymbol{U}_l^{\star T}\boldsymbol{x}^{(i)}\|=\|\boldsymbol{U}_l^{\star T}(\boldsymbol{U}_k^\star\boldsymbol{a}_i+\boldsymbol{e}_i)\|=\|\boldsymbol{U}_l^{\star T}\boldsymbol{e}_i\|\leq\|\boldsymbol{e}_i\|,\ \forall l\neq k.\tag{38}$$

This, together with (35) and $\|\boldsymbol{e}_i\|<(\sqrt{d}-2\sqrt{\log N})/2$, implies $\|\boldsymbol{U}_k^{\star T}\boldsymbol{x}_i\|\geq\|\boldsymbol{U}_l^{\star T}\boldsymbol{x}_i\|$ for all $l\neq k$. Therefore, we have $i\in C_k(\boldsymbol{\theta}^\star)$ due to (34). Therefore, we have $C_k^\star\subseteq C_k(\boldsymbol{\theta}^\star)$ for each $k\in[K]$. This, together with the fact that they respectively denote a partition of $[N]$, yields $C_k(\boldsymbol{\theta}^\star)=C_k^\star$ for each $k\in[K]$. Now, we compute

$$f(\boldsymbol{\theta}^\star)=\sum_{k=1}^K\sum_{i\in C_k^\star}\|\boldsymbol{U}_k^{\star T}\boldsymbol{x}^{(i)}\|^2=\sum_{k=1}^K\sum_{i\in C_k^\star}\|\boldsymbol{a}_i+\boldsymbol{U}_k^{\star T}\boldsymbol{e}_i\|^2$$

$$=\sum_{i=1}^N\|\boldsymbol{a}_i\|^2+2\sum_{k=1}^K\sum_{i\in C_k^\star}\langle\boldsymbol{a}_i,\boldsymbol{U}_k^{\star T}\boldsymbol{e}_i\rangle+\sum_{k=1}^K\sum_{i\in C_k^\star}\|\boldsymbol{U}_k^{\star T}\boldsymbol{e}_i\|^2.\tag{39}$$

Next, we compute

$$f(\hat{\boldsymbol{\theta}})=\sum_{k=1}^K\sum_{i\in C_k(\hat{\boldsymbol{\theta}})}\|\hat{\boldsymbol{U}}_k^T\boldsymbol{x}^{(i)}\|^2=\sum_{l=1}^K\sum_{k=1}^K\sum_{i\in C_k(\hat{\boldsymbol{\theta}})\cap C_l^\star}\|\hat{\boldsymbol{U}}_k^T(\boldsymbol{U}_l^\star\boldsymbol{a}_i+\boldsymbol{e}_i))\|^2$$

$$=\sum_{l=1}^K\sum_{k=1}^K\sum_{i\in C_k(\hat{\boldsymbol{\theta}})\cap C_l^\star}\left(\|\hat{\boldsymbol{U}}_k^T\boldsymbol{U}_l^\star\boldsymbol{a}_i\|^2+2\langle\boldsymbol{a}_i,\boldsymbol{U}_l^{\star T}\hat{\boldsymbol{U}}_k\hat{\boldsymbol{U}}_k^T\boldsymbol{e}_i\rangle\right)+\sum_{k=1}^K\sum_{i\in C_k(\hat{\boldsymbol{\theta}})}\|\hat{\boldsymbol{U}}_k^T\boldsymbol{e}_i\|^2.$$

This, together with $f(\hat{\boldsymbol{\theta}}) \geq f(\boldsymbol{\theta}^\star)$ and (39), yields

$$\sum_{i=1}^{N} \|\boldsymbol{a}_i\|^2 - \sum_{l=1}^{K} \sum_{k=1}^{K} \sum_{i \in C_k(\hat{\boldsymbol{\theta}}) \cap C_l^\star} \|\hat{\boldsymbol{U}}_k^T \boldsymbol{U}_l^\star \boldsymbol{a}_i\|^2 \leq \sum_{l=1}^{K} \sum_{k=1}^{K} \sum_{i \in C_k(\hat{\boldsymbol{\theta}}) \cap C_l^\star} 2\langle \boldsymbol{a}_i, \boldsymbol{U}_l^{\star T} \hat{\boldsymbol{U}}_k \hat{\boldsymbol{U}}_k^T \boldsymbol{e}_i \rangle +$$

$$\sum_{k=1}^{K} \sum_{i \in C_k(\hat{\boldsymbol{\theta}})} \|\hat{\boldsymbol{U}}_k^T \boldsymbol{e}_i\|^2 - 2\sum_{k=1}^{K} \sum_{i \in C_k^\star} \langle \boldsymbol{a}_i, \boldsymbol{U}_k^{\star T} \boldsymbol{e}_i \rangle - \sum_{k=1}^{K} \sum_{i \in C_k^\star} \|\boldsymbol{U}_k^{\star T} \boldsymbol{e}_i\|^2$$

$$\leq 4\delta \sum_{i=1}^{N} \|\boldsymbol{a}_i\| + N\delta^2 \leq 6\delta N\sqrt{d} + N\delta^2, \tag{40}$$

where the second inequality follows from $\|\boldsymbol{e}_i\| \leq \delta$ for all $i \in [N]$ and $\boldsymbol{U}_k^\star, \hat{\boldsymbol{U}}_k \in \mathcal{O}^{n \times d}$ for all $k \in [K]$, and the last inequality uses (35).

For ease of exposition, let $N_{kl} := |C_k(\hat{\boldsymbol{\theta}}) \cap C_l^\star|$. According to the pigeonhole principle, there exists a permutation $\pi : [K] \to [K]$ such that there exists $k \in [K]$ such that $N_{\pi(k)k} \geq N/K^2$. This, together with (40), yields

$$6\delta N\sqrt{d} + N\delta^2 \geq \sum_{i \in C_{\pi(k)}(\hat{\boldsymbol{\theta}}) \cap C_k^\star} \left( \|\boldsymbol{a}_i\|^2 - \|\hat{\boldsymbol{U}}_{\pi(k)}^T \boldsymbol{U}_k^\star \boldsymbol{a}_i\|^2 \right)$$

$$= \langle \boldsymbol{I} - \boldsymbol{U}_k^{\star T} \hat{\boldsymbol{U}}_{\pi(k)} \hat{\boldsymbol{U}}_{\pi(k)}^T \boldsymbol{U}_k^\star, \sum_{i \in C_{\pi(k)}(\hat{\boldsymbol{\theta}}) \cap C_k^\star} \boldsymbol{a}_i \boldsymbol{a}_i^T \rangle. \tag{41}$$

According to Lemma 6 and $N_{\pi(k)k} \geq N/K^2$, it holds with probability at least $1 - 2K^4 N^{-2}$ that

$$\left\| \frac{1}{N_{\pi(k)k}} \sum_{i \in C_{\pi(k)}(\hat{\boldsymbol{\theta}}) \cap C_k^\star} \boldsymbol{a}_i \boldsymbol{a}_i^T - \boldsymbol{I} \right\| \leq \frac{9(\sqrt{d} + \sqrt{\log(N_{\pi(k)k})})}{\sqrt{N_{\pi(k)k}}}.$$

This, together with the Weyl's inequality, yields

$$\lambda_{\min} \left( \sum_{i \in C_{\pi(k)}(\hat{\boldsymbol{\theta}}) \cap C_k^\star} \boldsymbol{a}_i \boldsymbol{a}_i^T \right) \geq N_{\pi(k)k} - 9\sqrt{N_{\pi(k)k}} \left( \sqrt{d} + \sqrt{\log(N_{\pi(k)k})} \right)$$

$$\geq \frac{N}{K^2} - \frac{9\sqrt{N}}{K} \left( \sqrt{d} + \sqrt{\log N} \right) \geq \frac{N}{2K^2},$$

where the second inequality follows from $N/K^2 \leq N_{\pi(k)k} \leq N$ and the last inequality is due to $\sqrt{N} \geq 18K(\sqrt{d} + \sqrt{\log N})$. Using this and Lemma 7, we obtain

$$\langle \boldsymbol{I} - \boldsymbol{U}_k^{\star T} \hat{\boldsymbol{U}}_{\pi(k)} \hat{\boldsymbol{U}}_{\pi(k)}^T \boldsymbol{U}_k^\star, \sum_{i \in C_{\pi(k)}(\hat{\boldsymbol{\theta}}) \cap C_k^\star} \boldsymbol{a}_i \boldsymbol{a}_i^T \rangle$$

$$\geq \lambda_{\min} \left( \sum_{i \in C_{\pi(k)}(\hat{\boldsymbol{\theta}}) \cap C_k^\star} \boldsymbol{a}_i \boldsymbol{a}_i^T \right) \mathrm{Tr} \left( \boldsymbol{I} - \boldsymbol{U}_k^{\star T} \hat{\boldsymbol{U}}_{\pi(k)} \hat{\boldsymbol{U}}_{\pi(k)}^T \boldsymbol{U}_k^\star \right)$$

$$\geq \frac{N}{2K^2} \mathrm{Tr} \left( \boldsymbol{I} - \boldsymbol{U}_k^{\star T} \hat{\boldsymbol{U}}_{\pi(k)} \hat{\boldsymbol{U}}_{\pi(k)}^T \boldsymbol{U}_k^\star \right).$$

This, together with (41), implies

$$\mathrm{Tr} \left( \boldsymbol{I} - \boldsymbol{U}_k^{\star T} \hat{\boldsymbol{U}}_{\pi(k)} \hat{\boldsymbol{U}}_{\pi(k)}^T \boldsymbol{U}_k^\star \right) \leq 2K^2 \left( 6\delta\sqrt{d} + \delta^2 \right).$$

Using this and $[\boldsymbol{U}_1^\star, \ldots, \boldsymbol{U}_k^\star] \in \mathcal{O}^{n \times dK}$, we obtain

$$\sum_{l \neq k} \|\hat{\boldsymbol{U}}_{\pi(k)}^T \boldsymbol{U}_l^\star\|_F^2 = \mathrm{Tr} \left( \sum_{l \neq k} \hat{\boldsymbol{U}}_{\pi(k)}^T \boldsymbol{U}_l^\star \boldsymbol{U}_l^{\star T} \hat{\boldsymbol{U}}_{\pi(k)} \right) \leq \mathrm{Tr} \left( \boldsymbol{I} - \hat{\boldsymbol{U}}_{\pi(k)}^T \boldsymbol{U}_k^\star \boldsymbol{U}_k^{\star T} \hat{\boldsymbol{U}}_{\pi(k)} \right)$$

$$\leq 2K^2 \left( 6\delta\sqrt{d} + \delta^2 \right) \leq \frac{3d}{4}, \tag{42}$$

where the last inequality follows $\delta \leq \sqrt{d}/(24K^2)$. According to (40), we have

$$
6\delta N\sqrt{d} + N\delta^2 \geq \sum_{l \neq k}^{K} \sum_{i \in C_{\pi(k)}(\hat{\boldsymbol{\theta}}) \cap C_l^{\star}} \left( \|\boldsymbol{a}_i\|^2 - \|\hat{\boldsymbol{U}}_{\pi(k)}^T \boldsymbol{U}_l^{\star} \boldsymbol{a}_i\|^2 \right)
$$

$$
\geq \sum_{l \neq k}^{K} N_{\pi(k)l} \left( (\sqrt{d} - \alpha)^2 - \left( \|\hat{\boldsymbol{U}}_{\pi(k)}^T \boldsymbol{U}_l^{\star}\|_F + \alpha \right)^2 \right) \geq \frac{d}{8} \sum_{l \neq k}^{K} N_{\pi(k)l},
$$

where the second inequality uses (35) and (36), and the last inequality follows from $d \gtrsim \log N$. Therefore, we have for each $k \in [K]$,

$$
\sum_{l \neq k}^{K} N_{\pi(k)l} \leq \frac{48\delta N\sqrt{d} + 8\delta^2 N}{d} < 1,
$$

where the last inequality uses $\delta \lesssim \sqrt{d/N}$. This implies $N_{\pi(l)k} = 0$ for all $l \neq k$, and thus $C_{\pi(k)}(\hat{\boldsymbol{\theta}}) \subseteq C_k^{\star}$. Using the same argument, we can show that $C_{\pi(l)}(\hat{\boldsymbol{\theta}}) \subseteq C_l^{\star}$ for each $l \neq k$. Therefore, we have $C_{\pi(k)}(\hat{\boldsymbol{\theta}}) = C_k^{\star}$ for each $k \in [K]$. In particular, using the union bound yields event holds with probability at least $1 - 2K^2N^{-1}$. Based on the above optimal assignment, we can further show:

(i) Suppose that $N_k \geq d$ for each $k \in [K]$. This, together with (i) in Theorem 2 and $N_k \geq d$, yields (19).

(ii) Suppose that there exists $k \in [K]$ such that $N_k < d$. This, together with (ii) in Theorem 2 and $N_k \geq d$, yields (20).

Finally, applying the union bound yields the probability of these events. □

## C  EXPERIMENTAL SETUPS IN SECTION 2.2

In this section, we provide detailed setups for the experiments in Section 2.2. These experiments aim to validate the assumptions that real-world image data satisfies a mixture of low-rank Gaussians and that the DAE is parameterized according to (9). To begin, we show that $\nabla_{\boldsymbol{x}_t}\mathbb{E}[\boldsymbol{x}_0|\boldsymbol{x}_t]$ is of low rank when $p_{\text{data}}$ follows a mixture of low-rank Gaussians and $\sum_{k=1}^{K} d_k \leq n$, where $n$ is the ambient dimension of training samples.

### C.1  VERIFICATION OF MIXTURE OF LOW-RANK GAUSSIAN DATA DISTRIBUTION

In this subsection, we demonstrate that a mixture of low-rank Gaussians is a reasonable and insightful model for approximating real-world image data distribution. To begin, we show that $\nabla_{\boldsymbol{x}_t}\mathbb{E}[\boldsymbol{x}_0|\boldsymbol{x}_t]$ is of low rank when $p_{\text{data}}$ follows a mixture of low-rank Gaussians with $\sum_{k=1}^{K} d_k \leq n$, where $n$ is the dimension of training samples.

**Lemma 2.** *Suppose that the data distribution $p_{\text{data}}$ follows a mixture of low-rank Gaussian distributions as defined in Definition 1. For all $t \in [0,1]$, it holds that*

$$
\min_{k \in [K]} d_k \leq \text{rank}\left(\nabla_{\boldsymbol{x}_t}\mathbb{E}[\boldsymbol{x}_0|\boldsymbol{x}_t]\right) \leq \sum_{k=1}^{K} d_k. \tag{43}
$$

*Proof.* For ease of exposition, let

$$
h_k(\boldsymbol{x}_t) := \exp\left(\phi_t \|\boldsymbol{U}_k^{\star T}\boldsymbol{x}_t\|^2\right), \ \forall k \in [K].
$$

Obviously, we have

$$
\nabla h_k(\boldsymbol{x}_t) := 2\phi_t \exp\left(\phi_t\|\boldsymbol{U}_k^{\star T}\boldsymbol{x}_t\|^2\right) \boldsymbol{U}_k^{\star}\boldsymbol{U}_k^{\star T}\boldsymbol{x}_t = 2\phi_t h_k(\boldsymbol{x}_t)\boldsymbol{U}_k^{\star}\boldsymbol{U}_k^{\star T}\boldsymbol{x}_t. \tag{44}
$$

According to Lemma 1, we have

$$\mathbb{E}[\boldsymbol{x}_0|\boldsymbol{x}_t] = \frac{s_t}{s_t^2 + \gamma_t^2} f(\boldsymbol{x}_t), \text{ where } f(\boldsymbol{x}_t) := \frac{\sum_{k=1}^{K} \pi_k h_k(\boldsymbol{x}_t) \boldsymbol{U}_k^{\star} \boldsymbol{U}_k^{\star T} \boldsymbol{x}_t}{\sum_{k=1}^{K} \pi_k h_k(\boldsymbol{x}_t)}.$$

Then, we compute

$$\nabla_{\boldsymbol{x}_t} f(\boldsymbol{x}_t) = \frac{1}{\sum_{k=1}^{K} \pi_k h_k(\boldsymbol{x}_t)} \left( 2\phi_t \sum_{k=1}^{K} \pi_k h_k(\boldsymbol{x}_t) \boldsymbol{U}_k^{\star} \boldsymbol{U}_k^{\star T} \boldsymbol{x}_t \boldsymbol{x}_t^T \boldsymbol{U}_k^{\star} \boldsymbol{U}_k^{\star T} + \sum_{k=1}^{K} \pi_k h_k(\boldsymbol{x}_t) \boldsymbol{U}_k^{\star} \boldsymbol{U}_k^{\star T} \right)$$

$$- \frac{2\phi_t}{\left( \sum_{k=1}^{K} \pi_k h_k(\boldsymbol{x}_t) \right)^2} \left( \sum_{k=1}^{K} \pi_k h_k(\boldsymbol{x}_t) \boldsymbol{U}_k^{\star} \boldsymbol{U}_k^{\star T} \boldsymbol{x}_t \right) \left( \sum_{k=1}^{K} \pi_k h_k(\boldsymbol{x}_t) \boldsymbol{U}_k^{\star} \boldsymbol{U}_k^{\star T} \boldsymbol{x}_t \right)^T$$

$$= \frac{1}{\sum_{k=1}^{K} \pi_k h_k(\boldsymbol{x}_t)} \sum_{k=1}^{K} \pi_k h_k(\boldsymbol{x}_t) \left( 2\phi_t \boldsymbol{U}_k^{\star} \boldsymbol{U}_k^{\star T} \boldsymbol{x}_t \boldsymbol{x}_t^T + \boldsymbol{I} \right) \boldsymbol{U}_k^{\star} \boldsymbol{U}_k^{\star T} -$$

$$\frac{2\phi_t}{\left( \sum_{k=1}^{K} \pi_k h_k(\boldsymbol{x}_t) \right)^2} \left( \sum_{k=1}^{K} \pi_k h_k(\boldsymbol{x}_t) \boldsymbol{U}_k^{\star} \boldsymbol{U}_k^{\star T} \right) \boldsymbol{x}_t \boldsymbol{x}_t^T \left( \sum_{k=1}^{K} \pi_k h_k(\boldsymbol{x}_t) \boldsymbol{U}_k^{\star} \boldsymbol{U}_k^{\star T} \right).$$

This directly yields (43) for all $t \in [0, 1]$. □

Now, we conduct experiments to illustrate that diffusion models trained on real-world image datasets exhibit similar low-rank properties to those described in the above proposition. Provided that the DAE $\boldsymbol{x}_{\boldsymbol{\theta}}(\boldsymbol{x}_t, t)$ is applied to estimate $\mathbb{E}[\boldsymbol{x}_0|\boldsymbol{x}_t]$, we estimate the rank of the Jacobian of the DAE, i.e., $\nabla_{\boldsymbol{x}_t} \boldsymbol{x}_{\boldsymbol{\theta}}(\boldsymbol{x}_t, t)$, on the real-world data distribution, where $\boldsymbol{\theta}$ denotes the parameters of U-Net architecture trained on the real dataset. Also, this estimation is based on the findings in Luo (2022); Zhang et al. (2023) that under the training loss in Equation (6), the DAE $\boldsymbol{x}_{\boldsymbol{\theta}}(\boldsymbol{x}_t, t)$ converge to $\mathbb{E}[\boldsymbol{x}_0|\boldsymbol{x}_t]$ as the number of training samples increases on the real data. We evaluate the numerical rank of the Jacobian of the DAE on four different datasets: CIFAR-10 Krizhevsky et al. (2009), CelebA Liu et al. (2015), FFHQ Kazemi & Sullivan (2014) and AFHQ Choi et al. (2020), where the ambient dimension $n = 3072$ for all datasets.

Given a random initial noise $\boldsymbol{x}_1 \sim \mathcal{N}(\boldsymbol{0}, \boldsymbol{I}_n)$, diffusion models generate a sequence of images $\{\boldsymbol{x}_t\}$ according to the reverse SDE in Eq. (4). Along the sampling trajectory $\{\boldsymbol{x}_t\}$, we calculate the Jacobian $\nabla_{\boldsymbol{x}_t} \boldsymbol{x}_{\boldsymbol{\theta}}(\boldsymbol{x}_t, t)$ and compute its numerical rank via

$$\text{rank} (\nabla_{\boldsymbol{x}_t} \boldsymbol{x}_{\boldsymbol{\theta}}(\boldsymbol{x}_t, t)) := \arg\min \left\{ r \in [1, n] : \frac{\sum_{i=1}^{r} \sigma_i^2 (\nabla_{\boldsymbol{x}_t} \boldsymbol{x}_{\boldsymbol{\theta}}(\boldsymbol{x}_t, t))}{\sum_{i=1}^{n} \sigma_i^2 (\nabla_{\boldsymbol{x}_t} \boldsymbol{x}_{\boldsymbol{\theta}}(\boldsymbol{x}_t, t))} > \eta^2 \right\}. \quad (45)$$

In our experiments, we set $\eta = 0.99$. In the implementation, we utilize the Elucidating Diffusion Model (EDM) with the EDM noise scheduler Karras et al. (2022) and DDPM++ architecture Song et al. (2020). Moreover, we employ an 18-step Heun's solver for sampling and present the results for 12 of these steps. For each dataset, we random sample 15 initial noise $\boldsymbol{x}_1$, calculate the mean of $\text{rank}(\nabla_{\boldsymbol{x}_t} \boldsymbol{x}_{\boldsymbol{\theta}}(\boldsymbol{x}_t, t))$ along the trajectory $\{\boldsymbol{x}_t\}$, and plot ratio of the numerical rank over the ambient dimension against the signal-noise-ratio (SNR) $1/\sigma_t$ in Figure 3, where $\sigma_t$ is defined in Eq. (3).

## C.2 VERIFICATION OF LOW-RANK NETWORK PARAMETERIZATION

In this subsection, we empirically investigate the properties of U-Net architectures in diffusion models and validate the simplification of the network architecture to Eq. (9). Based on the results in Appendix C.1, we use a mixture of low-rank Gaussian distributions for experiments. Here, we set $K = 2$, $n = 48$, $d_1 = d_2 = 6$, $\pi_1 = \pi_2 = 0.5$, and $N = 1000$ for the data model Definition 1. Moreover, We use the EDM noise scheduler and 18-step Heun's solver for both the U-Net and our proposed parameterization (9). To adapt the structure of the U-Net, we reshape each training sample into a 3D tensor with dimensions $4 \times 4 \times 3$, treating it as an image. Here, we use DDPM++ based diffusion models with a U-Net architecture. In each iteration, we randomly sampled a batch of image $\{\boldsymbol{x}^{(j)}\}_{j=1}^{\text{bs}} \subseteq \{\boldsymbol{x}^{(i)}\}_{i=1}^{N}$, along with a timestep $t^{(j)}$ and a noise $\boldsymbol{\epsilon}^{(j)}$ for each image in the batch

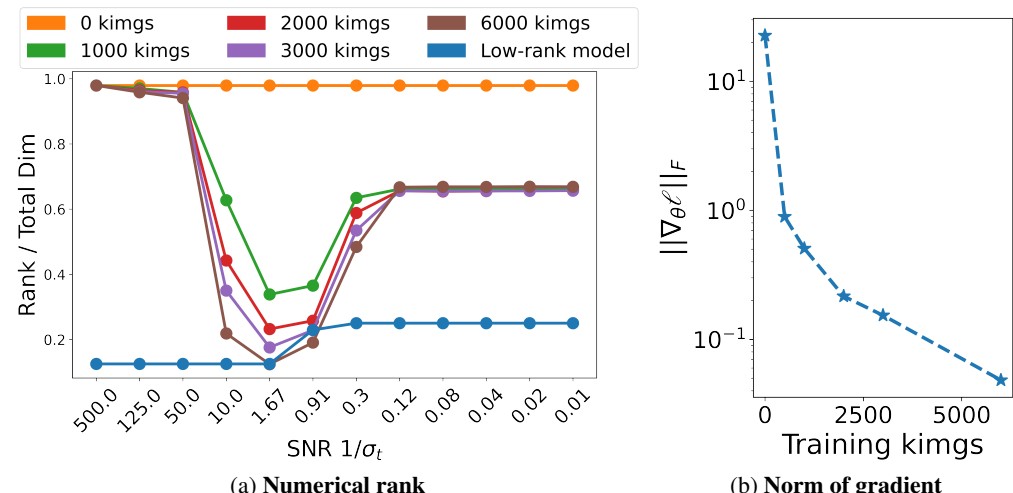

(a) **Numerical rank**  (b) **Norm of gradient**

Figure 6: (a) **Numerical rank of** $\nabla_{\boldsymbol{x}_t} \boldsymbol{x_\theta}(\boldsymbol{x}_t, t)$ **at all time of diffusion models.** Problem (6) is trained with the DAE $\boldsymbol{x_\theta}(\cdot, t)$ parameterized according to (9) and U-Net on the training samples generated by the mixture of low-rank Gaussian distribution. The $x$-axis is the SNR and the $y$-axis is the numerical rank of $\nabla_{\boldsymbol{x}_t} \boldsymbol{x_\theta}(\boldsymbol{x}_t, t)$ over the ambient dimension $n$, i.e., $\mathrm{rank}(\nabla_{\boldsymbol{x}_t} \boldsymbol{x_\theta}(\boldsymbol{x}_t, t))/n$. Here, *kimgs* denotes the number of samples used for training, which equals to training iterations times batch size of training samples. (b) **Convergence of gradient norm of the training loss**: The $x$-axis is kimgs (see Eq. (46)), and the $y$-axis is the gradient norm of the training loss.

to optimize the training loss $\ell(\boldsymbol{\theta})$. We define

$$\text{kimgs} = \text{bs} \times \frac{\text{training iterations}}{1000} \tag{46}$$

to represent the total samples used for training. Here, we pick up the specific model trained under 500 kimgs, 1000 kimgs, 2000 kimgs, and 6000 kimgs for evaluation, as shown in Figure 6(a).

We plot the numerical ranks of $\nabla_{\boldsymbol{x}_t} \boldsymbol{x_\theta}(\boldsymbol{x}_t, t)$ for both our proposed parameterization in (9) and for the U-Net architecture in Figure 3(b). According to Lemma 2, it holds that $6 \leq \mathrm{rank}(\nabla_{\boldsymbol{x}_t} \boldsymbol{x_\theta}(\boldsymbol{x}_t, t)) \leq 12$. This corresponds to the blue curve in Figure 3(b). To supplement our result in Figure 3(b), we further plot the numerical rank against SNR at different training iterations in Figure 6(a) and gradient norm of the objective against training iterations in Figure 6(b). We observe that with the training kimgs increases, the gradient for the U-Net $\|\nabla_{\boldsymbol{\theta}} \ell\|_F$ decrease smaller than $10^{-1}$ and the rank ratio of $\nabla_{\boldsymbol{x}_t} \boldsymbol{x_\theta}(\boldsymbol{x}_t, t)$ trained from U-Net gradually be close to the rank ratio from the low-rank model in the middle of the SNR ($[0.91, 10.0]$).

## D  EXPERIMENTAL SETUPS IN SECTION 4

We use a CPU to optimize Problem (6) for the setting in Appendix D.1. For the settings in Appendix D.2 and Appendix D.3, we employ a single A40 GPU with 48 GB memory to optimize Problem (6).

### D.1  LEARNING THE MOLRG DISTRIBUTION WITH THE THEORETICAL PARAMETERZATION

Here, we present the stochastic gradient descent (SGD) algorithm for solving Problem (6) as follows:

Now, we specify how to choose the parameters of the SGD in our implementation. We divide the time interval $[0, 1]$ into 64 time steps. When $K = 1$, we set the learning rate $\eta = 10^{-4}$, batch size $M = 128N_k$, and number of iterations $J = 10^4$. When $K = 2$, we set the learning rate $\eta = 2 \times 10^{-5}$, batch size $M = 1024$, number of iterations $J = 10^5$. In particular, when $K = 2$, we use the following tailor-designed initialization $\boldsymbol{\theta}^0 = \{\boldsymbol{U}_k^0\}$ to improve the convergence of the SGD:

$$\boldsymbol{U}_k^0 = \boldsymbol{U}_k^\star + 0.2\boldsymbol{\Delta}, \ k \in \{1, 2\}, \tag{47}$$

---

**Algorithm 1** SGD for optimizing the training loss (6)

---

**Input:** Training samples $\{\boldsymbol{x}^{(i)}\}_{i=1}^N$
**for** $j = 0, 1, 2, \ldots, J$ **do**
    Randomly select $\{(i_m, t_m)\}_{m=1}^M$, where $i_m \in [N]$ and $t_m \in (0, 1)$ and a noise $\boldsymbol{\epsilon} \sim \mathcal{N}(\boldsymbol{0}, \boldsymbol{I})$
    Take a gradient step

$$\boldsymbol{\theta}^{j+1} \leftarrow \boldsymbol{\theta}^j - \frac{\eta}{M} \sum_{m \in [M]} \nabla_{\boldsymbol{\theta}} \left\| \boldsymbol{x}_{\boldsymbol{\theta}^j}(s_{t_m} \boldsymbol{x}^{(i_m)} + \gamma_{t_m} \boldsymbol{\epsilon}, t_m) - \boldsymbol{x}^{(i_m)} \right\|^2$$

**end for**

---

where $\boldsymbol{\Delta} \sim \mathcal{N}(\boldsymbol{0}, \boldsymbol{I}_n)$. We calculate the success rate as follows. If the returned subspace basis matrices $\{\boldsymbol{U}_k\}_{k=1}^K$ satisfy

$$\frac{1}{K} \sum_{k=1}^K \|\boldsymbol{U}_{\Pi(k)} \boldsymbol{U}_{\Pi(k)}^T - \boldsymbol{U}_k^{\star} \boldsymbol{U}_k^{\star T}\| \leq 0.5$$

for some permutation $\Pi : [K] \to [K]$, it is considered successful.

### D.2 LEARNING THE MoLRG DISTRIBUTION WITH U-NET

we measure the generalization ability of U-Net via *generalization (GL) score* defined in Equation (48).

$$\text{GL score} = \frac{\mathcal{D}(\boldsymbol{x}_{\text{gen}}^{(i)})}{\mathcal{D}(\boldsymbol{x}_{\text{MoLRG}}^{(i)})}, \quad \mathcal{D}(\boldsymbol{x}^{(i)}) := \sum_{j=1}^N \min_{j \neq i} \|\boldsymbol{x}^{(i)} - \boldsymbol{x}^{(j)}\|, \tag{48}$$

where $\{\boldsymbol{x}_{\text{MoLRG}}^{(i)}\}_{i=1}^N$ are samples generated from the MoLRG distribution and $\{\boldsymbol{x}_{\text{gen}}^{(i)}\}_{i=1}^N$ are new samples generated by the trained U-Net. Intuitively, $\mathcal{D}(\boldsymbol{x}_{\text{gen}}^{(i)})$ reflects the uniformity of samples in the space: its value is small when the generated samples cluster around the training data, while the value is large when generated samples disperse in the entire space. Therefore, the trained diffsion model is in memorization regime when $D(\boldsymbol{x}_{\text{gen}}^{(i)}) \ll \mathcal{D}(\boldsymbol{x}_{\text{MoLRG}}^{(i)})$ and the GL score is close to 0, while it is in generalization regime when $D(\boldsymbol{x}_{\text{gen}}^{(i)}) \approx \mathcal{D}(\boldsymbol{x}_{\text{MoLRG}}^{(i)})$ and the GL score is close to 1.

In our implementation, we set the total dimension of MoLRG as $n = 48$ and the number of training samples $N_{\text{eval}} = 1000$. To train the U-Net, we use the stochastic gradient descent in Algorithm 1. We use DDPM++ architecture Song et al. (2021) for the U-Net and EDM Karras et al. (2022) noise scheduler. We set the learning rate $10^{-3}$, batch size 64, and number of iterations $J = 10^4$.

### D.3 LEARNING REAL-WORLD IMAGE DATA DISTRIBUTIONS WITH U-NET

According to Zhang et al. (2023), we define the generalization (GL) score on real-world image dataset as follows:

$$\text{GL score} := 1 - \mathbb{P}\left(\max_{i \in [N]} [\mathcal{M}_{\text{SSCD}}(\boldsymbol{x}, \boldsymbol{y}_i)] > 0.6\right). \tag{49}$$

Here, the SSCD similarity is first introduced in Pizzi et al. (2022) to measure the replication between image pair $(\boldsymbol{x}_1, \boldsymbol{x}_2)$, which is defined as follows:

$$\mathcal{M}_{\text{SSCD}}(\boldsymbol{x}_1, \boldsymbol{x}_2) = \frac{\text{SSCD}(\boldsymbol{x}_1) \cdot \text{SSCD}(\boldsymbol{x}_2)}{\|\text{SSCD}(\boldsymbol{x}_1)\|_2 \cdot \|\text{SSCD}(\boldsymbol{x}_2)\|_2}$$

where $\text{SSCD}(\cdot)$ represents a neural descriptor for copy detection of images. We empirically sample 10K initial noises to estimate the probability. Intuitively, GL score measures the dissimilarity between the generated sample $\boldsymbol{x}$ and all $N$ samples $\boldsymbol{y}_i$ from the training dataset $\{\boldsymbol{y}_i\}_{i=1}^N$.

To train diffusion models for real-world image datasets, we use the DDPM++ architecture Song et al. (2021) for the U-Net and variance preserving (VP) Song et al. (2021) noise scheduler. The U-Net is trained using the Adam optimizer Kingma & Ba (2014), a variant of SGD in Algorithm 1. We set the learning rate $\eta = 10^{-3}$, batch size $M = 512$, and the total number of iterations $10^5$.

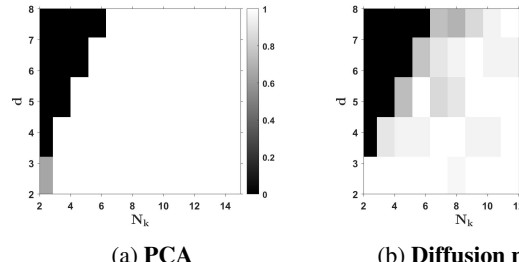

(a) **PCA**          (b) **Diffusion model**

Figure 7: **Phase transition of learning the MoLRG distribution when** $K = 3$**.** The $x$-axis is the number of training samples and $y$-axis is the dimension of subspaces. We apply a subspace clustering method and train diffusion models for solving Problems (18) and (6), visualizing the results in (a) and (b), respectively.

### D.4 CORRESPONDENCE BETWEEN LOW-DIMENSIONAL SUBSPACES AND IMAGE SEMANTICS

We denote the Jacobian of the DAE $\boldsymbol{x_\theta}(\boldsymbol{x}_t, t)$ by $\boldsymbol{J}_t := \nabla_{\boldsymbol{x}_t}\boldsymbol{x_\theta}(\boldsymbol{x}_t, t) \in \mathbb{R}^{n \times n}$ and let $\boldsymbol{J}_t = \boldsymbol{U\Sigma V}^T$ be an singular value decomposition (SVD) of $\boldsymbol{J}_t$, where $r = \text{rank}(\boldsymbol{J}_t)$, $\boldsymbol{U} = [\boldsymbol{u}_1, \cdots, \boldsymbol{u}_r] \in \mathcal{O}^{n \times r}$, $\boldsymbol{V} = [\boldsymbol{v}_1, \cdots, \boldsymbol{v}_r] \in \mathcal{O}^{n \times r}$, and $\boldsymbol{\Sigma} = \text{diag}(\sigma_1, \ldots, \sigma_r)$ with $\sigma_1 \geq \cdots \geq \sigma_r$ being the singular values. According to the results in Figure 3, it is observed that $\boldsymbol{J}_t$ is low rank, i.e., $r \ll n$. Now, we compute the first-order approximation of $\boldsymbol{x_\theta}(\boldsymbol{x}_t, t)$ along the direction of $\boldsymbol{v}_i \in \mathbb{R}^n$, where $\boldsymbol{v}_i$ is the $i$-th right singular vector of $\boldsymbol{J}_t$:

$$\boldsymbol{x_\theta}(\boldsymbol{x}_t + \alpha\boldsymbol{v}_i, t) \approx \boldsymbol{x_\theta}(\boldsymbol{x}_t, t) + \alpha\boldsymbol{J}_t\boldsymbol{v}_i = \boldsymbol{x_\theta}(\boldsymbol{x}_t, t) + \alpha\sigma_i\boldsymbol{u}_i,$$

where the last equality follows from $\boldsymbol{J}_t\boldsymbol{v}_i = \boldsymbol{U\Sigma V}^T\boldsymbol{v}_i = \alpha\sigma_i\boldsymbol{u}_i$. To validate the semantic meaning of the basis $\boldsymbol{v}_i$, we vary the value of $\alpha$ from negative to positive and visualize the resulting changes in the generated images. Figures 2, 8 and 9(a, c) illustrate some real examples.

In the experiments, we use a pre-trained diffusion denoising probabilistic model (DDPM) Ho et al. (2020) on the MetFaces dataset Karras et al. (2020). We randomly select an image $\boldsymbol{x}_0$ from this dataset and use the reverse process of the diffusion denoising implicit model (DDIM) Song et al. (2020) to generate $\boldsymbol{x}_t$ at $t = 0.7T$ (ablation studies for $t = 0.1T$ and $0.9T$ are shown in Figure 9(b)), where $T$ denote the total number of time steps. We respectively choose the changed direction as the leading right singular vectors $\boldsymbol{v}_1, \boldsymbol{v}_3, \boldsymbol{v}_4, \boldsymbol{v}_5, \boldsymbol{v}_6$ and use $\tilde{\boldsymbol{x}}_t = \boldsymbol{x}_t + \alpha\boldsymbol{v}_i$ to generate new images with $\alpha \in [-6, 6]$ shown in Figures 2, 8 and 9(a, c).

## E AUXILIARY RESULTS

First, we present a probabilistic result to prove Theorem 2, which provides an optimal estimate of the small singular values of a matrix with i.i.d. Gaussian entries. This lemma is proved in (Rudelson & Vershynin, 2009, Theorem 1.1).

**Lemma 3.** *Let $\boldsymbol{A}$ be an $m \times n$ random matrix, where $m \geq n$, whose elements are independent copies of a subgaussian random variable with mean zero and unit variance. It holds for every $\varepsilon > 0$ that*

$$\mathbb{P}\left(\sigma_{\min}(\boldsymbol{A}) \geq \varepsilon(\sqrt{m} - \sqrt{n-1})\right) \geq 1 - (c_1\varepsilon)^{m-n+1} - \exp\left(-c_2 m\right),$$

*where $c_1, c_2 > 0$ are constants depending polynomially only on the subgaussian moment.*

Next, we present a probabilistic bound on the deviation of the norm of weighted sum of squared Gaussian random variables from its mean. This is a direct extension of (Vershynin, 2018, Theorem 5.2.2).

**Lemma 4.** *Let $\boldsymbol{x} \sim \mathcal{N}(\boldsymbol{0}, \boldsymbol{I}_d)$ be a Gaussian random vector and $\lambda_1, \ldots, \lambda_d > 0$ be constants. It holds for any $t > 0$ that*

$$\mathbb{P}\left(\left|\sqrt{\sum_{i=1}^d \lambda_i^2 x_i^2} - \sqrt{\sum_{i=1}^d \lambda_i^2}\right| \geq t + 2\lambda_{\max}\right) \leq 2\exp\left(-\frac{t^2}{2\lambda_{\max}^2}\right), \tag{50}$$

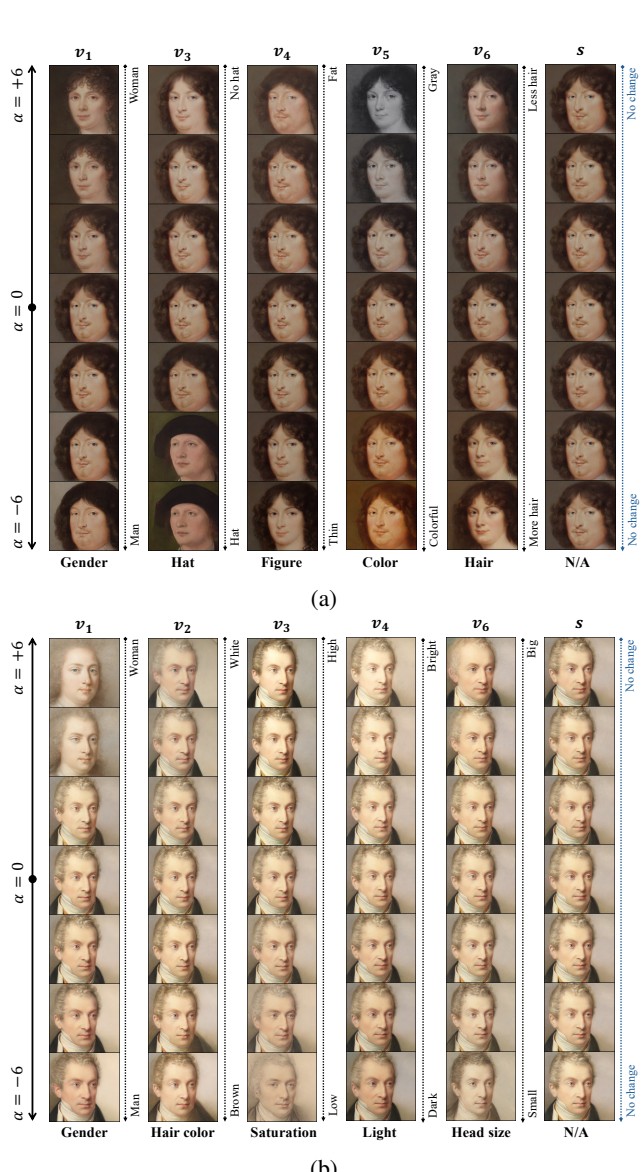

Figure 8: **Correspondence between the singular vectors of the Jacobian of the DAE and semantic image attributes.**

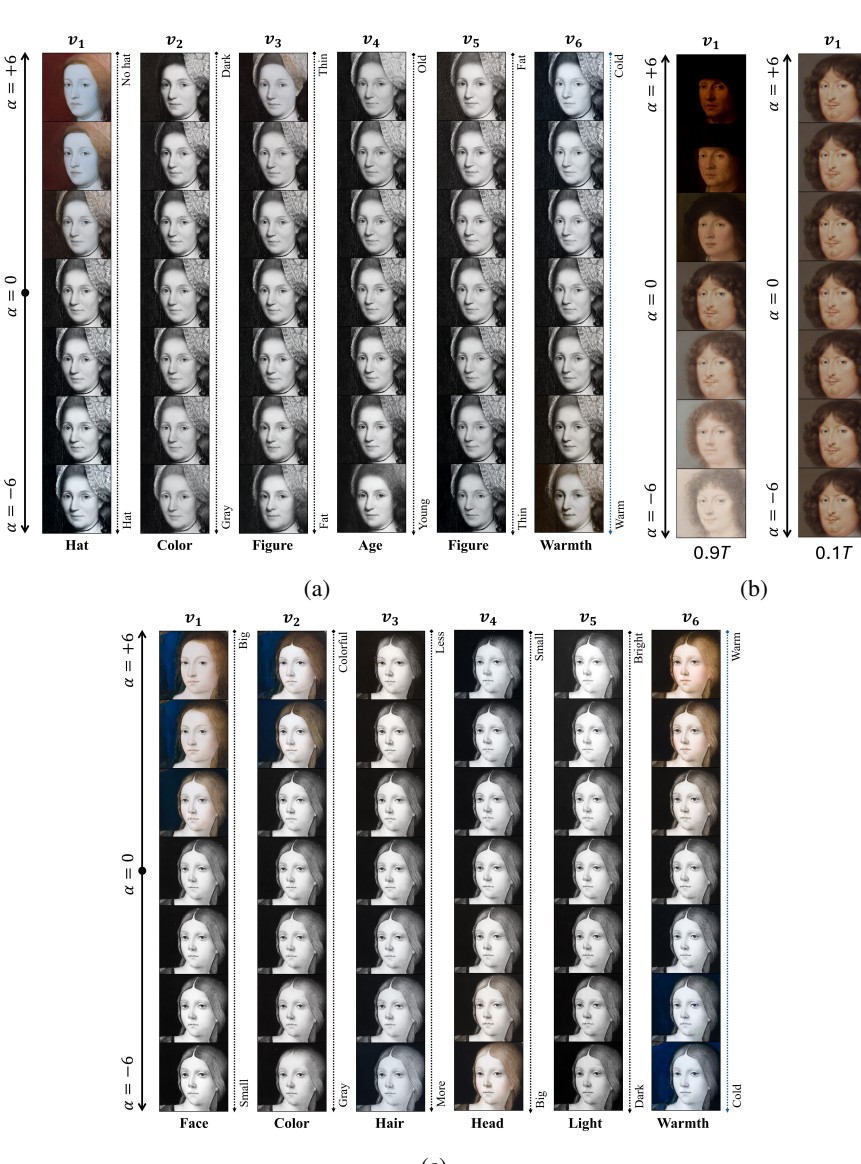

Figure 9: **Correspondence between the singular vectors of the Jacobian of the DAE and semantic image attributes.** (a,c) Additional examples when $t = 0.7T$. (b) Ablation studies when $t = 0.1T$ and $0.9T$.

*where $\lambda_{\max} = \max\{\lambda_i : i \in [d]\}$.*

Based on the above lemma, we can further show the following concentration inequalities to estimate the norm of the standard norm Gaussian random vector.

**Lemma 5.** *Suppose that $\boldsymbol{a}_i \overset{i.i.d.}{\sim} \mathcal{N}(\boldsymbol{0}, \boldsymbol{I}_d)$ is a Gaussian random vector for each $i \in [N]$. The following statements hold:*
*(i) It holds for all $i \in [N]$ with probability at least $1 - N^{-1}$ that*

$$\left| \|\boldsymbol{a}_i\| - \sqrt{d} \right| \le 2\sqrt{\log N} + 2. \tag{51}$$

*(ii) Let $\boldsymbol{V} \in \boldsymbol{O}^{n \times d}$ be given. For all $i \in C_k^\star$ and all $k \in [K]$, it holds with probability at least $1 - 2N^{-1}$ that*

$$\left| \|\boldsymbol{V}^T \boldsymbol{U}_k^\star \boldsymbol{a}_i\| - \|\boldsymbol{V}^T \boldsymbol{U}_k^\star\|_F \right| \le 2\sqrt{\log N} + 2. \tag{52}$$

*Proof.* (i) Applying Lemma 4 to $\boldsymbol{a}_i \sim \mathcal{N}(\boldsymbol{0}, \boldsymbol{I}_d)$, together with setting $t = 2\sqrt{\log N}$ and $\lambda_j = 1$ for all $j \in [d]$, yields

$$\mathbb{P}\left( \left| \|\boldsymbol{a}_i\| - \sqrt{d} \right| \ge 2\sqrt{\log N} + 2 \right) \le 2N^{-2}.$$

This, together with the union bound, yields that (51) holds with probability $1 - N^{-1}$.

(ii) Let $\boldsymbol{V}^T \boldsymbol{U}_k^\star = \boldsymbol{P} \boldsymbol{\Sigma} \boldsymbol{Q}^T$ be a singular value decomposition of $\boldsymbol{V}^T \boldsymbol{U}_k^\star$, where $\boldsymbol{\Sigma} \in \mathbb{R}^{d \times d}$ with the diagonal elements $0 \le \sigma_d \le \dots \sigma_1 \le 1$ being the singular values of $\boldsymbol{V}^T \boldsymbol{U}_k^\star$ and $\boldsymbol{P}, \boldsymbol{Q} \in \mathcal{O}^d$. This, together with the orthogonal invariance of the Gaussian distribution, yields

$$\|\boldsymbol{V}^T \boldsymbol{U}_k^\star \boldsymbol{a}_i\| = \|\boldsymbol{\Sigma} \boldsymbol{Q}^T \boldsymbol{a}_i\| \overset{d}{=} \|\boldsymbol{\Sigma} \boldsymbol{a}_i\| = \sqrt{\sum_{j=1}^d \sigma_j^2 a_{ij}^2}. \tag{53}$$

Using Lemma 4 with setting $t = 2\sigma_1 \sqrt{\log N}$ and $\lambda_j = \sigma_j \le 1$ for all $j$ yields

$$\mathbb{P}\left( \left| \|\boldsymbol{V}^T \boldsymbol{U}_k^\star \boldsymbol{a}_i\| - \|\boldsymbol{V}^T \boldsymbol{U}_k^\star\|_F \right| \ge \sigma_1 \alpha \right) = \mathbb{P}\left( \left| \sqrt{\sum_{j=1}^d \sigma_j^2 a_{ij}^2} - \sqrt{\sum_{j=1}^d \sigma_j^2} \right| \ge \sigma_1 \alpha \right) \le 2N^{-2}.$$

This, together with $\sigma_1 \le 1$ and the union bound, yields (52). $\qquad \square$

Next, We present a spectral bound on the covariance estimation for the random vectors generated by the normal distribution.

**Lemma 6.** *Suppose that $\boldsymbol{a}_1, \dots, \boldsymbol{a}_N \in \mathbb{R}^d$ are i.i.d. standard normal random vectors, i.e., $\boldsymbol{a}_i \overset{i.i.d.}{\sim} \mathcal{N}(\boldsymbol{0}, \boldsymbol{I}_d)$ for all $i \in [N]$. Then, it holds with probability at least $1 - 2N^{-2}$ that*

$$\left\| \frac{1}{N} \sum_{i=1}^N \boldsymbol{a}_i \boldsymbol{a}_i^T - \boldsymbol{I}_d \right\| \le \frac{9(\sqrt{d} + \sqrt{\log N})}{\sqrt{N}}, \tag{54}$$

*Proof.* According to (Vershynin, 2018, Theorem 4.7.1), it holds that

$$\mathbb{P}\left( \left\| \frac{1}{N} \sum_{i=1}^N \boldsymbol{a}_i \boldsymbol{a}_i^T - \boldsymbol{I}_d \right\| \ge \frac{9(\sqrt{d} + \eta)}{\sqrt{N}} \right) \le 2 \exp\left( -2\eta^2 \right),$$

where $\eta > 0$. Plugging $\eta = \sqrt{\log N}$ into the above inequality yields

$$\mathbb{P}\left( \left\| \frac{1}{N} \sum_{i=1}^N \boldsymbol{a}_i \boldsymbol{a}_i^T - \boldsymbol{I}_d \right\| \ge \frac{9(\sqrt{d} + \sqrt{\log N})}{\sqrt{N}} \right) \le 2N^{-2}.$$

This directly implies (54). $\qquad \square$

**Lemma 7.** *Let* $\boldsymbol{A}, \boldsymbol{B} \in \mathbb{R}^{n \times n}$ *be positive semi-definite matrices. Then, it holds that*

$$\langle \boldsymbol{A}, \boldsymbol{B} \rangle \geq \lambda_{\min}(\boldsymbol{A}) \mathrm{Tr}(\boldsymbol{B}). \tag{55}$$

*Proof.* Let $\boldsymbol{U}\boldsymbol{\Lambda}\boldsymbol{U}^T = \boldsymbol{A}$ be an eigenvalue decompositon of $\boldsymbol{A}$, where $\boldsymbol{U} \in \mathcal{O}^n$ and $\boldsymbol{\Sigma} = \mathrm{diag}(\lambda_1, \ldots, \lambda_n)$ is a diagonal matrix with diagonal entries $\lambda_1 \geq \cdots \geq \lambda_n \geq 0$ being the eigenvalues. Then, we compute

$$\langle \boldsymbol{A}, \boldsymbol{B} \rangle = \langle \boldsymbol{U}\boldsymbol{\Lambda}\boldsymbol{U}^T, \boldsymbol{B} \rangle = \langle \boldsymbol{\Lambda}, \boldsymbol{U}\boldsymbol{B}\boldsymbol{U}^T \rangle \geq \lambda_{\min}(\boldsymbol{A})\mathrm{Tr}(\boldsymbol{U}\boldsymbol{B}\boldsymbol{U}^T) = \lambda_{\min}(\boldsymbol{A})\mathrm{Tr}(\boldsymbol{B}),$$

where the inequality follows from $\lambda_i \geq 0$ for all $i \in [N]$ and $\boldsymbol{B}$ is a positive semidefinite matrix. $\square$

