# OpenReview forum: "Diffusion Models Learn Low-Dimensional Distributions via Subspace Clustering"
_ICLR.cc/2025/Conference — ICLR 2025 Conference Withdrawn Submission_

### Official Review · Reviewer_U6rt · 2024-10-28

**Soundness:** 2
**Presentation:** 3
**Contribution:** 1
**Rating:** 3
**Confidence:** 2

**Summary:**

Diffusion models are the dominant class of image generation models. They can effectively learn the underlying image distribution during training, despite the high dimensionality of the data. This paper offers a theoretical modeling of the image distribution using a mixture of low-rank Gaussian. With this model, the authors attempt to explain the training dynamics and failure points of training with small datasets, as well as offer several insights.

**Strengths:**

- The attempt to model and quantify the training dynamics and data size requirements for diffusion models is noteworthy and possibly impactful.
- The motivation behind the MoLRG is intuitive and easy to follow.
- The paper is well written, making both the math and the figures accessible.

**Weaknesses:**

- I believe the authors should highlight the paper's contribution. Whether the modeling is justified or not, I believe the paper should contain a meaningful takeaway message. For example, some numerical relationship between the denoiser's jacobian's rank and the number of samples required.
- The experiments conducted do not provide sufficient convincing evidence that the chosen modeling is fitting. Moreover, it is unclear what supports the application of the conclusion following Theorem 4 to real data.
- I am uncertain why modeling the DAE as a mixture of zero-meaned Gaussian is justified. Assuming that the data is a union of linear subspaces, using low-rank Gaussians is reasonable yet the subspaces do not necessarily coincide at the origin. Could the authors please shed more light on this choice?
- The use of the principle components of a Denoiser's jacobian for semantic exploration has been explored in previous work, namely [1]. Also, the connection between the proposed MoLRG and the semantic correspondence of the DAE's jacobian's principle component was unclear.

[1] Hila Manor & Tomer Michaeli (2024). On the Posterior Distribution in Denoising: Application to Uncertainty Quantification. In ICLR 2024.

**Questions:**

- Is it possible to show some meaningful bounds on the gap between the modeling using MoLRG and the real data distribution?
- Are the findings in the paper relevant specifically for diffusion models or for generative models in general? If the findings are general, it would strengthen the paper to present similar results across different generative methods. Otherwise, It would be interesting to shed some light on the difference in data size requirements of different generative modeling techniques.

---

### Official Review · Reviewer_sKVx · 2024-11-01

**Soundness:** 3
**Presentation:** 3
**Contribution:** 2
**Rating:** 6
**Confidence:** 4

**Summary:**

This work is about understanding the generalization capabilities of diffusion models from a theoretical perspective. The main assumption is that the true underlying data distribution can be approximated by a mixture of low rank Gaussians. Following this assumption, the authors propose an ideal parametrization for denoising networks of diffusion models. Combined with a few more approximations (e.g. hard max counterpart of weight assignments), they propose: 1) denoising is reduced to a sub-space clustering problem (in theorem 3) , and 2) the error in approximating the true sub-spaces is related to sub-space dimensionality and number of data points in training samples (theorem 4).

**Strengths:**

1. The paper raises an important question regarding generalization requirements in diffusion models. So the paper is significant in the sense that it is about a crucial question.
2. Extending previous theoretical work from mixture of full rank Gaussians to low rank Gaussians is an important and valuable step, since a mixture of low rank Gaussians is a better model of natural images (and generally real world structured data).
3. Additionally, showing empirical results in both toy and real data is a plus.
4. Finally, the paper is written clearly and is fairly easy to follow.

**Weaknesses:**

1. The main weakness of the paper is that the main limitation of the analysis is not acknowledged clearly enough. This limitation is due to the main assumption of the work: image data lies on a fixed union of low dimensional sub-spaces.  Although better than a full-rank Gaussian model, this is still a very crude approximation. As cited by the authors, image data lies on a union of **non-linear** manifolds, which cannot necessarily be accurately approximated by a union of linear manifolds (sub-spaces). The nice linear relationships between N and d will not hold as soon as you have non-linear manifolds. For example, for k =1, you would need N=2 to get a perfect estimate of a one dimensional subspace with a linearity assumption. But as soon as you have a non-linear manifold, depending on the degree of non-linearity you would need larger N.
I suggest the presentation should be modified to reflect the limitation of the analysis. Otherwise, the way the paper is written now, sets up the reader for disappointment, as the results do not extend to the real image data as claimed by the authors. Nevertheless, I think the results are valuable regardless of the simplistic assumption. They just need to be upfront about the assumptions.
2. Modeling image distribution with a union of sub-spaces is an old idea in signal processing that goes back to wavelet thresholding and later compressive sensing literature. The paper would benefit from citing major papers where these ideas originated. Importantly, the optimal solutions under this kind of assumption has been a very active topic in that area. It would be interesting to see how these results connect to that literature.
3. Another assumption made is to approximate the weights with a hard-max operation. This is of course a very hard assumption that results in a lot of error for high noise levels (when noisy image is far from the union of subspaces). Importantly, this assumption simplifies the posterior mean too much, which is counter productive when the goal is to explain diffusion models (where you have large levels of noise).
4. There seems to be a confusion in the experimental results presented in Figure 5. As a reader, I expect the experimental result to support the theory. However, the generalization result rely on a generalization score that is defined in the appendix. The results shows that the generalization as defined in eq 48 is related to N_k/d_k in a sensible way. However, it does not support nor refutes the results presented in the 4 theories (main results). So there is a divide between the theoretical results and the empirical results which is supposed to support the theory.

**Questions:**

1. Figure 2 shows changes in the image as a function of moving in the direction of top singular vectors of the Jacobian of the denoiser. The semantic labels are strange, because there are multiple features changing in each column but only one is chosen as a description. The complex variations of course is a reflection of the fact that the manifolds are not linear and it is not trivial to separate the features. Overall it is not clear that this figure is trying to convey. The actual effect is not consistent with the description.
2. In figure 3, for real image datasets, there is a jump at the very low SNR for estimated rank from 3 of the dataset. What is causing this strange behavior? Similarly for the toy data, the behavior of UNet seems pretty strange and non monotonic. Is there an intuition or explanation for that?
3. In line 364, it's stated that the assumption $U_k^TU_l = 0$ follows from the observation of disjoint union of manifold. a) This is another too strong assumption about images. We know that images share features across different classes and images, so it is not natural to assume this orthogonality between the subspaces. b) Even if we assume this orthogonality, why does this follow from the observation that the manifolds are disjoint? That refers to the support to be non-overlapping. In the general case, when the manifolds are not mean zero, they can have many directions in parallel and still be disjoint. c) Finally, if this is an assumption you are making, it is in consistent with the mixture of Gaussian models you show in figure 1, where your orthogonality assumption does not hold.
4. In figure 4,  it seems like diffusion models are performing worse than the simple PCA models in terms of separating the success and failure cases. Is it clear why?
5. I suggest eq 48 and 49 be moved to the main text because they are used to generate figures in the main text.

---

### Official Review · Reviewer_kQCq · 2024-11-03

**Soundness:** 3
**Presentation:** 3
**Contribution:** 2
**Rating:** 3
**Confidence:** 3

**Summary:**

The paper suggests that diffusion models can circumvent curse of dimensionality by clustering to fit the intrinsic dimension which is in general much lower than ambient dimension. This is a valuable insight and the authors provided a solid theoretical analysis to support it in a special case. However, the paper suffers from two draw backs: (1) the study of diffusions along subspaces and their study using PCA techniques is not a completely novel ideas; (2) the study focuses narrowly on finite sums of low-rank Gaussians centered at the origin (up to a noise), effectively making the search space finite dimensional.

**Strengths:**

The paper argues that instead of generating a full ranked diffusion, the diffusion model only need to solve a subproblem corresponding to the intrinsic dimension of the data distribution. This is a very convincing idea. In addition, in the case of MoLRG (mixture of low rank Gaussians). the authors mathematically proves the problem is equivalent to a PCA optimization and established rigorous guarantees of the optimization quality.

**Weaknesses:**

Weaknesses:
1. It is not a new idea to study behaviors of diffusion models, or generative models in general, along subspaces spanned by leading eigenvectors of the Jacobian. I cannot produce a complete bibliography in this area, but the following papers may be relevant:
- Subspace Diffusion Generative Models, Jing et al., ECCV 2022 , which developed a new diffusion model that restricts the flow vector field to a linear subspace whose dimension shrinks as t->0
- The Geometry of Deep Generative Image Models and its Applications, Wang & Ponce, ICLR 2021, whose experiments focused on GANs rather than diffusion models, but revealed that generative models identifies top eigenspaces of the Jacobian, which capture important perceptually relevant changes.
2. The study in this paper assumes the distribution is a sum of finitely many Gaussians centered at 0 with added noises (and even assumed later that they have the same weights). This is an oversimplification as under the current assumption, the task becomes an optimization of finitely matrices. The real-life diffusion is much more complicated as it tries to identify,  instead of linear subspaces,  submanifolds or equilvalently a tangent linear subspace at each point, which is an infinite dimensional task. How would the results change for data lying on low dimensional non-linear manifold? I guess the complexity of the task would depend on how fast the tangent space change among nearby points, which is quantified as curvatures of the manifold. It would be interesting to see discussions addressing this aspect.
3. This contrast is clearly demonstrated by the current paper's own experiments: under the simplified assumption, only a handful of samples (equal to the rank of the matrix) are needed for decent inference quality while for realistic data thousands of samples are needed ( Figure 5a vs Figure 5b ). While it is true the ranking of difficulty among different datasets coincide with that of intrinsic dimension, it should be recognized that a great amount of new details exist in data with higher instrinsic dimensions, and model quality a priori cannot be naively summarized by transferring the analysis on low-rank Gaussians in Theorems 1-4. I think it would be interesting to have more analysis on whether local intrinsic dimension are the only features that need to be handled in real work datasets, in particular, whether the shape and smoothness of the distribution also play a role.

**Questions:**

My questions are listed in the weakness section above. I would encourage the authors to analyze more carefully the transferability of claims from MoLRG at the origin toward MoLRG at every basepoint.

---

### Note · Authors · 2024-11-23

I have read and agree with the venue's withdrawal policy on behalf of myself and my co-authors.